# Towards viscous debris flows simulation using DualSPHysics v5.2 : Internal behaviour of viscous flows and mixtures

Suzanne Lapillonne[1], Georgios Fourtakas[2], Vincent Richefeu[3], Guillaume Piton[1], and Guillaume Chambon[1]

[1]Univ. Grenoble Alpes, INRAE, CNRS, IRD, Grenoble INP, IGE, Grenoble, France
[2]School of Engineering, University of Manchester, Manchester, M13 9PL, UK
[3]Univ. Grenoble Alpes, 3SR, Gieres, France

**Correspondence:** Suzanne Lapillonne (lapillonne.s@gmail.com)

**Abstract.** This paper investigates the accuracy of a solid-fluid model using the SPH software DualSPHysics v5.2 coupled with ProjectChrono for viscous debris flow modelling. It focuses on different validation steps of the method, both for pure fluid and a mixture of fluid and boulders to build reliability of the model to prepare for the simulation of a simplified debris flow. First, velocity profiles, free surface shape and velocity of surges of a viscous fluid are validated against well documented experimental data. It is, to the best of our knowledge, one of the few validations of the SPH approach for very viscous flows near the creeping threshold ($Re \approx 1$). Secondly, the influence on the macroscopic viscosity of the introduction of granular elements in a viscous fluid is studied against a semi-empirical formula. Finally, the method is applied to simplified $2D$ debris flow surges with field-like features. Surges modelled in this paper are composed of a viscous Newtonian fluid and poly-disperse boulders. The flow of surges of different concentrations is studied and Froude numbers of real field measurements are retrieved. Such complex models are shown to be relevant to the study of debris flow dynamics.

## 1 Introduction

Debris flows are rapid flows saturated with non-plastic debris in a steep channel (Hungr, 2005). These fast flows yield suddenly, behaving as so-called surges, creating a granular front which has the potential to be very destructive, followed by a viscous matrix engulfing granular material. Debris flows evolve naturally in steep, erosion-prone catchments and can be triggered by abundant runoff (Bel, 2017; Bernard et al., 2025), landslides (Iverson, 1997; Recking et al., 2013), or snow and ice melt (Recking et al., 2013). Once initiated, the flow propagates downstream, often recruiting material from the channel through entrainment (Simoni et al., 2020; Reid et al., 2016). In the European Alps, the material transported by the flow usually comes from weathering of mountain hill-slopes (Recking et al., 2013). This leads to the presence of viscous dominated debris flow, with high clay content in the interstitial fluid and granular materials of a wide verity of sizes, from sand to boulders (Coussot et al., 1998). Efforts of the scientific community in the debris-flow field are growing in every direction, some focusing on triggering conditions and predictability (e.g. Bel et al., 2017; Berti et al., 2012, among others), others on impact forces with hazard mitigation infrastructures (e.g. Albaba et al., 2015; Poudyal et al., 2019, among others) and the latter focusing on internal dynamics within the flow (e.g. Laigle and Labbe, 2017; Leonardi et al., 2014, among others). To better mitigate

the risks associated with debris flows, there is a need to better understand the internal mechanics of debris flows and their interactions with infrastructures. Due to their destructive nature, it is not possible to measure directly within the flow, and experimental small-scale facilities cannot fully replicate how debris flow propagate and maintain their dynamics, due to scale effects. This leads to the scientific community turning towards numerical methods as a relevant mode of action to describe internal dynamics of debris-flow motion.

Strategies to model debris flow phenomena are adapted to the scale of interest. First, event scale models replicate the debris flow run-out and spreading at a large scale, spanning over the entire reach. Such models are typically used as an input for the design of hazard maps on a debris fan. Event scale models rely of depth-averaged methods. Depth-averaged models can either be single phase or multiphase. The first consider the debris flow material as one homogenised phase. The rheology of the phase differs between models, some using a Voellmy rheology (Christen et al., 2010; Pastor et al., 2009; Pirulli et al., 2005, among others), while others use a Herschel-Bulkley or Bingham rhelogy (Laigle and Coussot, 1997; Gibson et al., 2021, among others). Such models are typically user-friendly, and thus popular for engineering applications. RAMMS (Christen et al., 2010) and HEC-RAS 2D (Gibson et al., 2021) are two of the common examples used in engineering for debris flow modelling. However, because these models are single phase, there is no change in density or rheological parameters along the flow path and duration, thus there is a lack of representation of local physical processes which impact the macroscopic behaviour of the flow. To improve this aspect, multi-phase depth-averaged models have started to bloom in the past decade. They typically represent the debris flow material with one phase representing the liquid contribution and one or two phases representing the solid contribution. They are then able to represent local effects within the phase interactions, such as dilatancy, pore-fluid pressure, entrainment, de-watering or washing over deposition. The most common models in this method are D-CLAW (George and Iverson, 2014), r.avaflow 2.0 (Mergili and Pudasaini, 2014), Titan2D (Pastor et al., 2018) and the recently improved version of RAMMS (Meyrat et al., 2022). However, depth-averaged models, by nature, are only adapted to large scale representation of the flow as they intrinsically neglect the momentum changes in depth, and thus, when dealing with the mechanical changes inside the debris flow material during the flow, they become limited. The large boulders that are transported by debris flows cannot be explicitly described in such models.

Conversely, surge-scale-or models with smaller spatial domains aim at representing the physics of debris flow dynamics at a mesoscopic and microscopic scale to better understand internal motion of the material and its interaction with infrastructures. They rely on explicit models where the velocity gradients in the depth of the flow are solved (e.g. Albaba et al., 2015; Laigle and Labbe, 2017; Chambon et al., 2011; Mitsoulis et al., 1993; Roquet and Saramito, 2003; Kong et al., 2021; Leonardi et al., 2014). For such a strategy, models can be classified into three types : (*i*) fluid models, modelling the debris flow material as a continuum of fluid with adapted rheology, (*ii*) granular models which model the debris flow material as an ensemble of grains and (*iii*) coupled methods which represent both phases.

(*i*) fluid models represent debris flows as a continuous fluid with complex rheology (Chambon et al., 2011; Labbé, 2015; Mitsoulis et al., 1993; Rodriguez-Paz and Bonet, 2004; Roquet and Saramito, 2003). The rheology can be fitted to observation to recover macroscopic flow features. Time-independent rheological constitutive models have proven to be effective to model debris flows when the volumetric fractions of silts and clay is sufficient and local discontinuities, segregation and concentration

heterogeneity can be neglected (Ancey, 2003; Laigle and Labbe, 2017). As a first approximation, muddy debris-flows, which are mechanically driven by viscous motion, can be studied at the macroscopic scale by considering a homogenous flow of a Newtonian viscous fluid (Han et al., 2014) or of a more complex material such as a Herschel-Bulkley fluid (Coussot and Meunier, 1996; Chambon et al., 2011). Such methods are often able to reproduce experimental results (Laigle and Labbe, 2017; Chambon et al., 2011) and retrieve characteristics of the flow that are close to field reality, with Froude numbers ranging from 0.5 to 3 (Marchi et al., 2002; Lapillonne et al., 2023; McArdell et al., 2023). These methods perform well for studies of the interactions with obstacles or spreading onto a deposition area. However, they neglect or simplify the granular skeleton in the flow. While the granular phase may be represented rheologically, the design of such models lacks the input of solid-solid interactions of the granular content onto the macroscopic flow. Thus, they cannot explore the role of granular skeleton in debris flow motion (Kaitna et al., 2011), or conditions to barrier clogging (Piton et al., 2022).

(*ii*) Granular models solely represent the granular phase of the debris flow material (Albaba et al., 2015; Ceccato et al., 2018), omitting the fluid input or strongly simplifying it by adapting the force interactions between the grains. They mainly focus on the estimation of impact forces of the boulders by the flow (Albaba et al., 2015; Ceccato et al., 2018). These models are effective to study impact forces applied to an infrastructure, and general rearrangement of the granular skeleton in a granular flow, but the absence of resolved interstitial fluid makes their parallel with real debris flows difficult, especially when it comes to studying established flows. Due to their granular nature, they also tend to require very steep flowing conditions with highly supercritical Froude numbers (up to 7). Such steep sites exist but seldom have assets located nearby or mitigation infrastructures, since those are typically rather located nearby alluvial fans or valley bottom where the slopes are milder.

Finally, (*iii*) coupled methods have grown in the past years thanks scientific and technological advances. These few hybrid methods are promising : resulting forces are computed with all inputs, and the interplay between large grains and the interstitial fluid is represented. They are able to explore larger scientific questions but require thorough validation. Kong et al. (2021) modelled debris flows using a coupled solver with Computational Fluid Dynamics (CFD) for the fluid phase and Discrete Element Method (DEM) for the granular phase. The model was validated against analytical predictions, field measurements and experimental results for the impact load on the barrier (Li et al., 2021). They were able to investigate the motion of the debris flow material over a wide range of Froude numbers (0.6–8.7). However, due to the numerical complexity of CFD-DEM solvers, the computational cost of such models is relatively high. Due to this technical inconvenience, they relied on initializing the velocity profiles of the flow as uniform close to the obstacle (1 m upstream). While the no-slip boundary condition lead to a retrieval of a velocity gradient, the distance at which the uniform velocity profile is initialised has unknown effect on the results. Thus, the model renders interesting results for barrier impact investigation but could not be applied to study debris flow motion on a larger time-scale. Other examples of models studying the gravity-driven free surface flow of a mixture of large solid particles and fluid can also show the high potential of CFD-DEM to fully represent mechanics of such mixtures (Peng et al., 2021; Robb et al., 2016).

Canelas et al. (2017) and Trujillo-Vela et al. (2020) employed the Smoothed Particle Hydrodynamics (SPH) framework with DEM. Canelas et al. (2017) studied the flow of granular material and water, which could bear some resemblance to granular debris flows. Their study contributed to understand how this flow interacts with a filtering dam structure. However,

they used a low viscosity fluid to represent the interstitial fluid which distinguishes it from the alpine debris flows that this paper addresses. Trujillo-Vela et al. (2020) added a soil phase with a Drucker-Prager criterion to study the run-out distance of a mixture of soil, water and a few boulders (3). They were able to validate their pure fluid behaviours accurately against analytical derivations. However, their low boulder concentration does not allow for granular macroscopic effects to be retrieved, such as the cristallization of a granular skeleton or the viscosity change due to the introduction of boulders.

Finally, Leonardi et al. (2014) investigated a Lattice Boltzman solver coupled with DEM for debris flows modelling. They successfully validated both the internal dynamics of the fluid during motion, the macroscopic viscosity response of the introduction of boulders in the fluid and the macroscopic response of granular collapse (Leonardi et al., 2014) and flow fronts (Leonardi et al., 2015). This method is able to investigate the complexity of the debris flow surge but has a significantly high sophistication and computational cost.

In this work, we build upon these attempts towards a model for viscous debris flow modelling using a coupled solid-fluid method. The work presented in this paper aims to conduct a similar approach to Leonardi et al. (2014) on a model using SPH for the fluid phase, in order to obtain a fully validated model that has a lower sophistication, so that the computational time and repeatability become more sensible for engineering and exploratory uses. Debris flows considered in this paper fall under the viscous debris flow category (Takahashi, 2014), common in the French Alps. They are modelled with a simplified composition : a solid phase capturing the bigger sediment fraction and an interstitial fluid that is typically highly viscous (Coussot and Meunier, 1996; Bardou, 2002). A Lagrangian method was chosen to be more appropriate as such methods deal straightforwardly with interfaces both at the free-surface (especially for the steep free-surface at the front of debris flow surges) and with the granular material. In this paper, the SPH method was chosen because of its simplicity to describe the free-surface and the interactions with solid elements. The DualSPHysics solver (Domínguez et al., 2021) is used as it is a widely available open-source software and already has the implementation of ProjectChrono (Tasora et al., 2016), a solid-solid solver based on the DVI method which will model collisions of boulders. The solver was also chosen for practical arguments : it is highly documented online, is GPU- and CPU-accelerated, is opensource, has relatively low computational time (compared to traditional eulerian methods and DEM) and has a growing community. This makes it a great candidate to be used more widely in studies of mass movements. However, the DualSPHysics solver is rarely used to model very viscous flows and it has not be validated against experimental or field observations. This papers seeks to check the reliability of this tool by confronting it against validation data. Future users will then better know the capability and limitations of their solver.

In this paper, debris flow material is simplified as a viscous Newtonian fluid loaded with granular elements for which explicit collisions are computed. There are no macroscopic computations representing collision through rheology and the fluid phase is purely Newtonian. First, the paper presents the methodology of the construction of the model. Secondly, we investigate the performance of the software to represent viscous surges both regarding the macroscopic and internal behaviour of the flow. We reproduce numerically experimental data from Freydier et al. (2017) and match velocity profiles and free surface shapes to the experimental results. Such precise validation of SPH on laminar viscous fronts are rare, but necessary to move forward in modelling of viscous geophysical and industrial flows. Thirdly, the macroscopic viscosity of a liquid-solid mixtures for different volumetric fraction is validated against the Krieger-Dougherty semi-empirical equation (Krieger and Dougherty,

1959). Finally, we draw from the fact that the method is able to both represent slow laminar flow dominated by the viscous regime and yield accurate changes in the macroscopic viscosity when the addition of boulders is explored to investigate a boulder-laden flow surge. In this last section, we present exploratory results to model a surge scale 2D debris-flow (in width) with an SPH based hybrid model, investigating three different solid concentrations and their influence on the macroscopic behaviour of the surge.

## 2   Materials and methods

### 2.1   SPH Modelling method

Smoothed Particles Hydrodynamics is a computational fluid dynamics method based on the lagrangian framework. In SPH, the continuous domain is discretized into numerical nodes ('particles'), which are points of known information. Typical properties of the continuum (e.g. velocity, density, … ) are associated to each of these points. The SPH method relies on the resolution of the Navier Stokes equation via interpolation onto these nodes. Particles interact with each other in a defined neighbourhood, named a smoothing kernel, by resolving the Navier Stokes equation system.

#### 2.1.1   Governing Equations

In SPH, the Navier-Stokes equations for a weakly-compressible fluid are used. In their Lagrangian form, they can be written as the following system (Liu and Liu, 2003):

$$\frac{D\rho}{Dt} = -\rho \nabla \cdot \underline{u} \tag{1}$$

$$\frac{D\underline{u}}{Dt} = \frac{1}{\rho}\nabla P + \frac{1}{\rho}\nabla \cdot \underline{\underline{\tau}} + \underline{g} \tag{2}$$

where $\rho$ is the density, $\underline{u}$ is the velocity, $P$ is the pressure, $\underline{\underline{\tau}}$ is the viscous stress tensor, $\underline{g}$ represents body forces, and $\frac{D}{Dt}$ stands for the material derivative. Note that we use as a notation $\underline{\cdots}$ for vectors and $\underline{\underline{\cdots}}$ for tensors. To close this system, an equation of state is used :

$$P = \frac{c_0^2 \rho_0}{\gamma}\left(\left(\frac{\rho}{\rho_0}\right)^{\gamma} - 1\right) \tag{3}$$

where $c_0$ is the numerical speed of sound, $\gamma = 7$ and $\rho_0$ is the reference density. This compressibility of SPH is purely artificial and required because of the explicit resolution of the Navier-Stokes equation.

For any point $\underline{r}$ in an ensemble $\Omega$ and a function $f : \mathbb{R}^3 \rightarrow \mathbb{R}$ continuous over $\Omega$, the following can always be written:

$$f(\underline{r}) = \int_{\Omega} \delta(\underline{r} - \underline{r}') f(\underline{r}') \, d\underline{r}' \tag{4}$$

where $\delta$ is the Dirac function. In continuous SPH, the key idea is to substitute the value of $f(\underline{r})$ by $\langle f(\underline{r}) \rangle$ as follows (Liu and Liu, 2003):

$$\langle f(\underline{r}) \rangle = \int_\Omega W_h(\underline{r} - \underline{r}') f(\underline{r}') \, \mathrm{d}\underline{r}' \tag{5}$$

where $W_h$ is weighted interpolation function of reaching length $h_W$ known as a smoothing kernel.

This integral procedure can then be discretized on numerical nodes. Applying this method to the governing equations (Eq. 1 and 2), the fluid is then discretized into so called SPH particles. These particles move along with the flow, representing a Lagrangian computation node with an assigned mass. Eq. 1 & 2 become, for a point $a$, computational node in a Newtonian fluid the following system:

$$\frac{\partial \rho_a}{\partial t} = \rho_a \sum_b \left( (\underline{u}_a - \underline{u}_b) \cdot \nabla W_{ab} \right) V_b + h_W \, c_0 \, \mathcal{D}_a \tag{6}$$

$$\frac{\partial \underline{u}_a}{\partial t} = -\sum_b m_b \left( \frac{P_a + P_b}{\rho_b \rho_a} \right) \nabla_a W_{ab} + \sum_b m_b \left( \frac{4\nu_0 \, \underline{r}_{ab} \cdot \nabla_a W_{ab}}{(\rho_a + \rho_b)(\underline{r}_{ab}^2 + \eta^2)} \right) \underline{u}_a + \underline{g} \tag{7}$$

$$\frac{\partial \underline{r}_a}{\partial t} = \underline{u}_a \tag{8}$$

where subscripts $a$ and $b$ refer to the interpolating and neighbouring particles respectively, $\underline{r}_i$ is the positional vector of particle $i$ and $\underline{r}_{ij} = \underline{r}_j - \underline{r}_i$ , $V_i$ the volume occupied by the particle, $m_i$ is the mass assigned to the numerical particle, $W_{ij}$ is the smoothing kernel, $\eta = 0.01 h_W$, to avoid singularity, $\mathcal{D}_i$ is a density diffusion term (see supplementary material) and $\nu_0$ the kinematic viscosity of the fluid. Eq. 7 is written for Newtonian fluids. By definition of how the solution is computed, there is an inherent "smoothing error" to the solution. The smoothing kernel used is the quintic Wendland (1995) kernel:

$$W(r, h_W) = \alpha_D \left( 1 - \frac{1}{2} \frac{r}{h_W} \right)^4 \left( 2 \frac{r}{h_W} + 1 \right) \quad \text{for} \quad 0 < \frac{r}{h_W} < 2 \tag{9}$$

where $\alpha_D$ depends on the dimension of the model: $\alpha_{2D} = \frac{7}{4} \pi h_W^2$ and $\alpha_{3D} = \frac{21}{16} \pi h_W^3$. The value of the smoothing length $h_W$ is set through a coefficient relating $h_W$ and $d_p$ : $h_W = C_h \sqrt{2} d_p$ (in 2D).

In this paper, the boundary conditioning is done using the mDBC method (*modified Dynamic Boundary Condition*) from English et al. (2021). Further information about the equations in the method used in this paper (time-stepping (Courant condition coefficient set to 0.2), boundary conditions, Lind et al. (2012) shifting method and density diffusion methods) can be found in the supplementary material, section 1.

## 2.2  Collision algorithm

In the DualSPHysics software, the CHRONO library (Anitescu and Tasora, 2008) is coupled with the SPH solver to deal with complex solid-fluid interactions (Martínez-Estévez et al., 2023). The rigid bodies are considered as a subset of the SPH

particle. The core concept of CHRONO solver method is based on the differential variational inequality (DVI) approach (Pang and Stewart, 2008) which shares mathematical framework with non smooth contact dynamics (Moreau, 1977; Jean and Moreau, 1992; Radjai and Richefeu, 2009). In a nutshell, the interaction between bodies are simultaneously computed for all contacts in a unified manner by considering the transitions between local and global/generalised frames. In addition to Newton's law relations involving impulses instead of forces, some conditions need to be considered to satisfy the non-penetration of the bodies (Signorini condition) and the Coulomb friction through inequalities. This model has been extensively used for different problems relating to rigid body dynamics, especially granular material and complex mechanical systems such as rovers and wheeled and tracked vehicles. A review of validated test cases can be found in Tasora et al. (2016).

In the coupling, the forces arising on the fluid-driven object are computed and integrated in time as:

$$
\begin{aligned}
M\frac{dV}{dt} &= \sum_k m_k \underline{f}_k \\
I\frac{d\Omega}{dt} &= \sum_k m_k ((\underline{r}_k - \underline{R}_0) \times \underline{f}_k) \cdot \underline{z}
\end{aligned}
\tag{10}
$$

where the subscript $k$ denotes an SPH boundary particle with a force per unit mass $f_k$, $\underline{V}$ is the linear velocity, $M$ is the mass of the rigid body with $R_0$ the position of the centre of mass, $I$ the moment of inertia and $\Omega$ the rotational velocity at a position $\underline{r}_k$.

CHRONO solver performs sub-iterations within a SPH timestep to adjust the rigid body kinematics, and gives back to SPH solver an updated centre of mass with linear and angular velocities, respectively $\underline{V}$ and $\Omega$. The velocity of each particle boundary in SPH is updated as

$$
\underline{u}_k = \underline{V} + \Omega \underline{z} \times (\underline{r}_k - \underline{R}_0)
\tag{11}
$$

and performing a system update for fluid and boundary particles. For more information the reader is directed to Martínez-Estévez et al. (2023).

## 2.3 Physical properties

The aim of this paper is to model viscous laminar debris flow surges with Reynolds number close to the creeping flow limits ($Re \approx 0.1$). We consider 2D flows of surges over an inclined plate.

Surges are assumed to be composed of a non-uniform front followed by a uniform zone, where the free surface elevation $z_{\max}$ becomes parallel to the bed, $\partial_x z_{\max} = 0$, where $\underline{x}$ is the direction longitudinal to the flow.

Hunt (1994) proposed to study theoretically the velocity profiles in a viscous laminar front, applied to debris flow processes. Assuming a steady flow of a fully developed laminar flow of a Newtonian fluid down an incline in 2D, the Navier-Stokes equations yield in the uniform zone :

$$
\begin{aligned}
p &= \rho g (z_{\max} - z) \cos\theta \\
u_x &= \frac{\rho g}{2\mu} \left( z_{\max}^2 - (z - z_{\max})^2 \right) \sin\theta \\
\overline{||\underline{u}||} &= \frac{\rho g \sin\theta}{3\mu} z_{\max}^2
\end{aligned}
\tag{12}
$$

**Table 1.** Characteristics of the experiment, density $\rho$ , viscosity $\mu$ , slope angle $\theta$ , conveyor belt velocity $u_b$ , Reynolds number $Re$

| exp. ID | $\rho$ | $\mu$ | $\theta$ | $u_b$ | $Re$ |
|---|---|---|---|---|---|
| | [kg $\cdot$ m$^{-3}$] | [Pa $\cdot$ s] | [°] | [mm $\cdot$ s$^{-1}$] | [–] |
| ID$_1$ | 1383 | 5.6 | 15.3 | 75 | 0.34 |
| ID$_2$ | 1380 | 4.9 | 11.9 | 164 | 1.3 |

where $p$ is the pressure, $\rho$ is the density of the fluid, $g$ is the gravitational force, $\mu$ is the viscosity, $u$ is the velocity, with $\overline{\cdots}$ denoting an average, suffix $\cdots_x$ denoting the $\underline{x}$ component of the velocity, $||\cdots||$ is the magnitude operator while $\underline{\cdots}$ denotes a vector, $z$ is the distance from the incline (in the normal direction), $z_{\mathrm{max}}$ is the free surface elevation, $\theta$ is the angle of the incline, taking into account the boundary condition $u(0) = 0$.

## 2.4 Selected experimental setup

Freydier et al. (2017) present experimental data precisely measuring viscous surges, i.e. flowing material, close from the maximum depth, rushing over a dry bed. Using advanced velocity measurements techniques by image analysis of a seeded, transparent fluids enlighten with a longitudinal, vertical laser sheet, these experiments captured both the macroscopic behaviour of a viscous flow front - monitoring properties of the flow such as free surface elevation and front velocity - and internal dynamics within the flow front - measuring velocity profiles. The experimental setup is composed a conveyor belt tilted to a chosen angle, transparent side-walls, and a wall upstream the conveyor belt forcing a flow front to form steadily. This experimental setup is thoroughly described in Freydier et al. (2017) and Chambon et al. (2009). Fluids used are transparent mixtures of glucose and water allowing to measure accurate velocity fields and free surface profiles. Freydier (2017) showed that these flows have a viscosity independent from strain and strain-rate, varying with concentration of glucose in the mixture. High definition, high velocity images at different location of the flow and complete velocimetry within the surge provide a full description of both the free surface shape and the velocity fields within the flow.

The characteristics of the experiments used in this current work for numerical validation of the software are shown in Table 1.

## 2.5 Rheology of neutrally buoyant mixtures

One of the key questions of debris flow modelling is the accurate representation of the interactions between the fluid phase and the granular content. Presence of non-Brownian particles in the flow increases overall viscosity which then impacts the flow front velocity and flow height (Coussot and Ancey, 2022). Thus, accurately representing the rheological effects of the presence of solid particles in the mixture is essential to ensure the performance of *DualSPHysics* to model debris flow events.

In this section, we study the macroscopic viscosity of a hard disks suspension in a 2D Poiseuille flow against known semi-empirical equations. The macroscopic viscosity is studied as a function of the solid fraction (aerial fraction $\phi$).

Rheology of suspensions is a thoroughly investigated field. A review of the different approaches, challenges and rheological models can be found in (Guazzelli and Pouliquen, 2018). Theoretical approaches draw from the pioneering work by Einstein (1906), which relates the particle fraction $\phi_D$ where $D$ denotes the dimension of the problem to the apparent viscosity linearly. This development of non-Brownian physics is only valid for very dilute regimes ($\phi_D < 0.05$) and was extended to any dimension $D$ by Brady (1983) such that:

$$\mu_{eq} = \mu_0 \left( 1 + \frac{D+2}{2} \phi_D \right)$$ (13)

where $\mu_{eq}$ is the macroscopic viscosity of the mixture and $\mu_0$ is the viscosity of the interstitial fluid.

At larger volumetric fractions, the effect of neighbouring particles becomes significant and the coupled interactions are expected to induce a viscosity contribution of $O(\phi^2)$ and even $O(\phi^3)$ for concentrated suspension. However, the precise description of the rheology of such mixture remains difficult (Guazzelli and Pouliquen, 2018). Numerous attempts at describing this problem for higher volumetric fraction are known in 3D. They usually aim at recovering the Einstein equation at low concentration while describing the diverging the viscosity at high volumetric fractions (Guazzelli and Pouliquen, 2018). Among them, the Krieger and Dougherty (1959) model is one of the most widely used semi-empirical formulas to predict macroscopic viscosity from solid concentration. It describes the behaviour of the mixture as a power law with two fitting parameters : $\phi_m$ and $p$.

$$\frac{\mu_{eq}}{\mu_0} = \left( 1 - \frac{\phi}{\phi_m} \right)^{-p\phi_m}$$ (14)

where $\phi_m$ is the maximal flowing fraction, with typical values between 0.58 and 0.65 (Guazzelli and Pouliquen, 2018).

In order to recover the Einstein law for diluted suspension, the exponent $p$ is often considered to be equal to $p = 2.5$. This value has shown especially accurate results for diluted flows, but has shown to be overestimated for concentrated suspensions (Guazzelli and Pouliquen, 2018).

## 3   Results

### 3.1   Viscous Newtonian surges

In this section, the goal is to investigate the macroscopic and internal features of a surge of a viscous Newtonian fluid against the experiments of Freydier et al. (2017) presented in section 2.4

### 3.1.1   Numerical setup

The numerical setup reproduces the experimental setup as a 2D flow front (Fig. 1), with flow features of Table 1. A surface of simulated fluid is generated onto an inclined conveyor belt. The inclination of the plane is represented by an inclined gravity vector in order to have an horizontal flow, following the slope $\theta$. The fluid is contained by an upstream wall. Surface of the experiment has been set up so that the wall is further than the distance required to reach uniform height shown in (Freydier

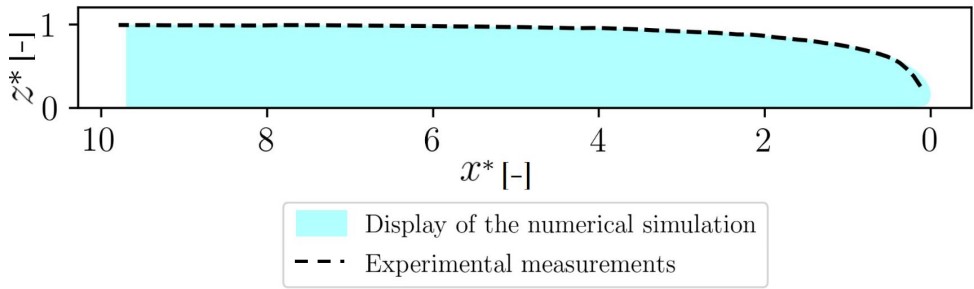

**Figure 1.** Display of the overall flow for simulated flow with SPH and free surface elevation of the experimental measurements for $ID_1$ with $d_p = 0.001m$ and $C_h = 1.0$, $x^* = x/h_{\max}$ and $z^* = z/h_{\max}$, where $h_{\max}$ is the theoretical maximal flow height in the uniform zone

et al., 2017). The flow height is initialized at the theoretical height of the uniform zone (from Eq. 12). At the initial time-step, the flow is set up to be a rectangle of length $L = 0.27m$, and height $h = 0.017m$ for $ID_1$ and $L = 0.4m$ and $h = 0.03m$ for experiment $ID_2$. The conveyor belt is simulated by a moving bed, moving towards the upstream direction and is implemented in a longitudinally periodic domain, as if infinite, to avoid any discontinuity on the boundary. It initiates a backwards motion, using a ramp to reach the velocity $u_b$. This ramp was put in place in order to avoid instabilities caused by the sudden infinite acceleration of an instantaneous motion.

A convergence study is performed on the results. For each case, five values of kernel coefficients $C_h$ are tested (tested values for $C_h$ :1, 1.3, 1.5, 1.8, and 2.2). Kernel coefficient are a measure of the ratio between smoothing length and particle spacing ($h_W = C_h\sqrt{2}d_p$ in 2D). Convergence with respect to the particle spacing distance $d_p$ is investigated for each value of $C_h$ (tested values for $d_p$ : $0.001m$, $0.005m$, and $0.00025m$) following (Quinlan et al., 2006). The initial particle spacing is chosen as such that $h/d_p > 15$ i.e. at least 15 SPH particles describe the flow in the uniform zone for any of the simulations.

The validation of the behaviour of the flow will be regarded as the accuracy to represent both macroscopic and internal features. The bulk flow behaviour is studied through (*i*) the velocity of the flow front, (*ii*) the free-surface elevation in the uniform zone and (*iii*) the free-surface shape. The internal kinematics of the flow are studied through (*iv*) the velocity profiles at 3 different locations in the flow front (Fig. 3a) which correspond to the positions studied in Chambon et al. (2019).

### 3.1.2 Data processing

The numerical model is run for a sufficient period of time, typically more than $30s$, ensuring steady state has been reached (see Figures S2 and S3 in the supplementary material). With SPH, velocity of the continuum can only be estimated through the velocity of the particles in the discretized flow. With the movement of particles in the flow intrinsic to the SPH method, instantaneous measurements can be parasited by fluctuations in the positions of the particles within the sampling window. To avoid these instantaneous effects, the results are averaged over 5 seconds.

For all results, heights and lengths are normalized by $h_{\max}$, the theoretical flow height in the uniform zone, so that $h^* = h/h_{\max}$ and $x^* = x/h_{\max}$. The position in $x$ are defined relatively to the tip of the front. This theoretical flow height is derived from Eq. 12. The velocities are normalized by the velocity of the conveyor belt $u_b$ so that $u^* = u/u_b$.

### 3.1.3 Bulk features of the flow

In Fig. 2, errors on the macroscopic behaviours are plotted. Fig. 2a shows the error on the velocity of the flow front, defined as:

$$\zeta_f = \frac{|u_f|}{u_b} = |u_f^*| \tag{15}$$

where $u_f$ is the velocity of the front provided by the numerical model averaged over one second, taken as the derivative of the location of the toe of the front. As a reminder, because the setup is on a conveyor belt going at $u_b$, $u_f \to 0$.

Fig. 2b shows the relative error on the flow height in the uniform zone for which $\partial_x h = 0$, defined as:

$$\zeta_h = \frac{|h_u - h_{\max}|}{h_{\max}} = |h_u^* - 1| \tag{16}$$

where $h_u^*$ is the normalised height in the uniform zone, taken in the uniform zone with $x^* > 9$.

Fig. 2c shows error on the overall free-surface profile as a root mean square error on the free surface elevation:

$$\zeta_{fs} = \sqrt{\frac{1}{N_x} \sum_{i=0}^{N_x} \left(\frac{h^*(x_i) - h_{\text{ref}}^*(x_i)}{h_{\text{ref}}^*(x_i)}\right)^2} \tag{17}$$

where $h^*(x_i)$ is the normalized free surface elevation of the simulated flow at position $x_i$, $h_{ref}^*(x)$ is its analogous normalized free surface elevation provided by the experimental data and $N_x$ is the number of points along $x$ on the front shape measurements provided by Freydier (2017).

Both experiments $ID_1$ and $ID_2$ are tested (see Table 1). In Figure 2, full symbols show the convergence study for experiment $ID_1$ and void symbols show the convergence study for experiment $ID_2$. Overall, the convergence is shown for both experiments and for the three features of interest. The numerical solution converges to less than $1\%$ of the theoretical value for $u_f$, see Fig. 2a, less than $5\%$ for $h_{\max}$, see Fig. 2b, and less than $10\%$ on the overall shape of the flow front, see Fig. 2c. The smoothing coefficient has a drastic influence on the prediction of the model: $C_h$ drives not only the numerical computation but the threshold of the method to detect the free-surface Domínguez et al. (2021). In applications where the free surface is very variable, as in this study, the value of the smoothing coefficient has a high impact on the overall performance of the model. A systematic study of the influence of such a coefficient is a relevant method to rule out any error from this choice.

### 3.1.4 Internal dynamics of the flows

Comparison between velocity profiles given by the numerical model and the profiles given by Freydier et al. (2017) at three different locations are studied for exp. $ID_1$. The same features were monitored for exp. $ID_2$ with similar performance (see

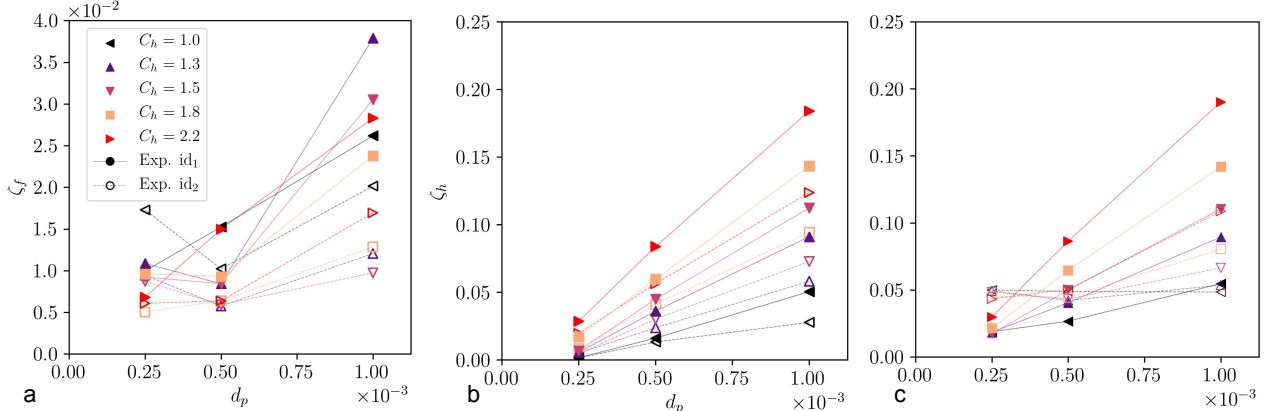

**Figure 2.** Convergence study on macroscopic features of the flow, $d_p$ in $m$ : a) Error on the flow front velocity as defined in Eq. 15, b)Error on the free surface elevation in the uniform zone as defined in Eq. 16, c) Root mean squared error on the surface shape for different resolutions and smoothing coefficient as defined in Eq. 17. Full symbols represent results of experiment $ID_1$, while void symbols represent results of the experiment $ID_2$.

Figure S1 in the supplementary data). In Fig. 3a, the three locations are highlighted, with the exact value of the positions shown in Table 1.

In Fig. 3b, c and d, the velocity profiles for $C_h = 1.8$ are plotted for each location and resolution. The overall behaviour of the flow is very well reproduced for all resolutions. Profiles are well within the error bars of the experimental data. Discrepancies on the height of the flow for $x_f^* = 0.76$ are due to the experimental measurements: close to the free-surface, access to the velocity of the flow becomes difficult (Freydier, 2017). Even near the front, the shape of the velocity profile is accurately reproducing the dynamics of the surge.

A global criterion to characterize the convergence, for each of the location $i$ is defined as the root mean squared error of the velocity profile along x :

$$\zeta_u = \sqrt{\frac{1}{N_h}\sum_{j=0}^{N_h}(u^*(h_j) - u^*_{\text{ref}}(h_i))^2} \tag{18}$$

where $u^*(h_j)$ is the normalized longitudinal velocity of the simulated flow at elevation $h_j$ in the flow, $u^*_{ref}(h_j)$ is its analogous normalized longitudinal velocity provided by the experimental data and $N_h$ is the number of points in the vertical direction,

which depends on the position $i$.

In Fig. 3e, f and g, the error $\zeta_{u,j}$ is plotted for each $(d_p, C_h)$. Convergence is shown for each smoothing coefficient. The convergence is similar regardless of the choice of a smoothing coefficient $C_h$, and overall, the error becomes smaller than $5\%$ when using the finest resolution. Considering the precision of the experimental measurements, this is very satisfactory.

To have a good agrement between both accuracy of macroscopic and internal features of the flow and computational time,

the smoothing coefficient $C_h = 1.8$ is chosen in the rest of the study.

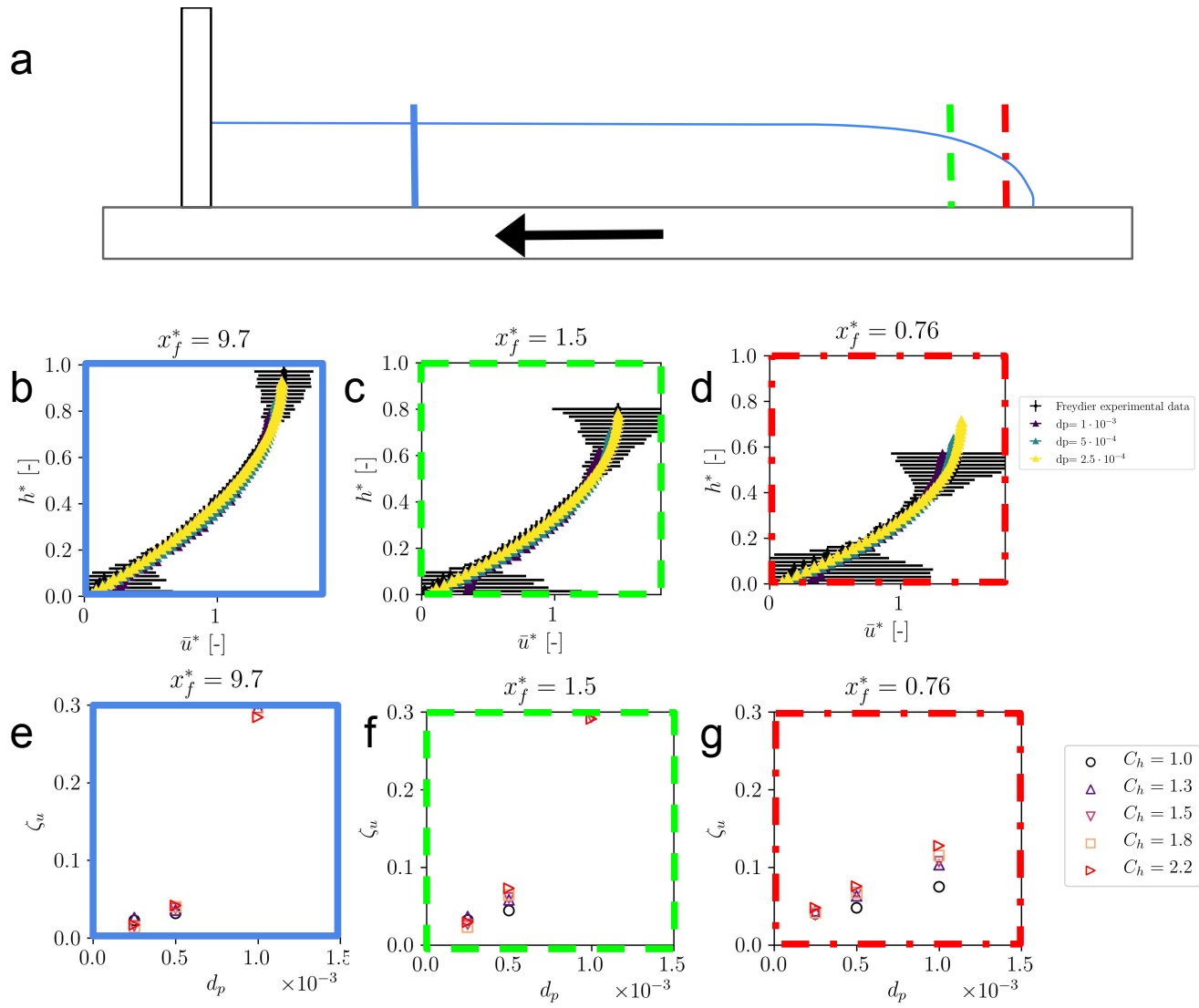

**Figure 3.** Study of the internal dynamics of the flow a) positions of the velocity profiles of interest: three different locations in the front : $x_f^* = 0.76$ and $x_f^* = 1.5$ being close to the front tip, where $\partial_x z$ is significant, and $x_f^* = 9.7$ in the uniform zone, b) c) and d) Velocity profiles of the numerical experiment plotted onto velocity profile of the experimental data by Freydier et al. (2017) for $C_h = 1.8$, convergence in $d_p$ is shown, e), f) and g) Root mean squared error as defined in Eq. 18 on the velocity profiles at the three positions showing the convergence study of the model in both $d_p$ and $C_h$, $d_p$ in m.

## 3.2 Macroscopic viscosity of mixtures

Following the validation of the highly viscous fluid in `DualSPHysics`, there is a missing piece to develop a more accurate representation of a 2D debris flow model. Indeed, the macroscopic viscosity changes due to the presence of solid particles in the flow. This needs to be correctly restituted by the model so that we can explore macroscopic behaviours of the debris flow model.

To test for the behaviour of a mixture, a 2D loaded Poiseuille test is performed. A fluid is generated between two horizontal plates, separated by a distance $B$, to which a pressure gradient is applied along the longitudinal direction. The pressure gradient is forced using a modified gravity in the $\underline{x}$ direction. The geometry is described in Fig. 4 where the boundaries are periodic along the $\underline{x}$ direction. The test is done using different concentration of monodisperse disks of radius $r$ randomly positioned in the channel. Each object has a no-slip condition with the fluid and is neutrally buoyant. The overall macroscopic viscosity is retrieved from the maximal velocity (Guyon et al., 2012) :

$$\mu_{eq} = \frac{B^2}{8u_{x,max}} \left| \frac{dp}{dx} \right| \tag{19}$$

The maximal velocity along $\underline{x}$, $u_{x,max}$, is taken as the average of the velocities along $\underline{x}$ of the fluid particles with positions at the center of the flume $\pm 1cm$ between $0.8$s and $1$s, once stationarity is reached (see Fig. S4).

The experimental designs are summarised in Table 2.

**Table 2.** Summary of the designs of the numerical experiments, note that for concentrations above 0.4, $d_p/r = 4.5\%$

| $\rho$ $[\mathrm{kg\,m^{-3}}]$ | $\|\mathbf{g}\| = \frac{1}{\rho}\|\nabla P\|$ $[\mathrm{m\,s^{-2}}]$ | $\mu_0$ $[\mathrm{m^2\,s^{-1}}]$ | $L$ $[\mathrm{m}]$ | $B$ $[\mathrm{m}]$ | $\phi$ $[-]$ | $r_{max}/L$ $[-]$ | $d_p/r$ $[-]$ | $C_h$ $[-]$ | Collision algorithm |
|---|---|---|---|---|---|---|---|---|---|
| 1000 | 0.8 | 0.467 | 1.0 | 0.6 | [0.05, 0.5] | 2% | 8.5% or 4.5% | 1.8 | CHRONO |

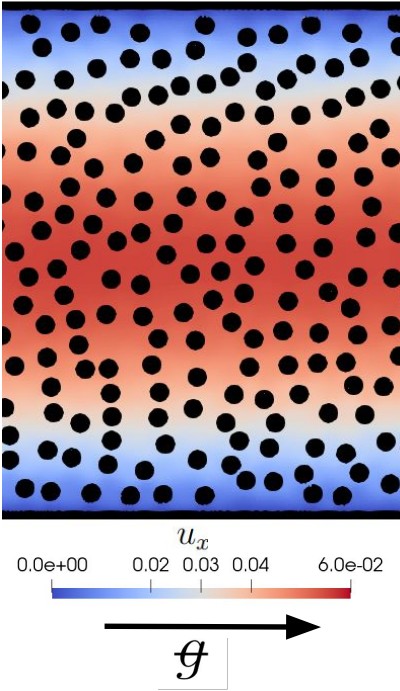

**Figure 4.** Experimental setup for the loaded Poiseuille case, the pressure gradient is introduced through $\boldsymbol{g}$, the colorbar represents velocity in the longitudinal direction

In Fig. 5, the comparison between the numerical results of the macroscopic viscosity against solid concentration and the empirical model is plotted. Eq. 14 is fitted for both $p$ and $\phi_m$. The black lines represent the uncertainty due to the spatial averaging.

The overall shape of the results does match the typical shape of this predictive model and the results can be fitted with precision. The root mean squared error of the fit is evaluated as :

$$\text{RMSE} = \sqrt{\frac{1}{N_s} \sum_{i=0}^{N_s} \left( \frac{\mu_{eq}}{\mu_0} - \frac{\mu_{est}}{\mu_0} \right)^2} \tag{20}$$

where $N_s$ is the number of tested concentrations and $\mu_{est}$ is the estimated equivalent viscosity using the parameters of the fit.

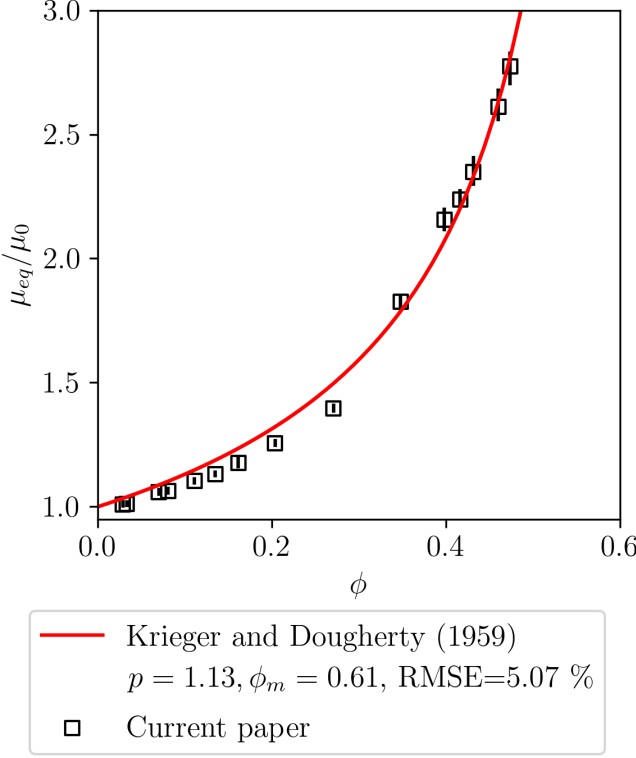

**Figure 5.** Evolution of the macroscopic viscosity with solid concentration, with the numerical data and its error bars.

The macroscopic response to the concentration of solid particles is satisfactory : the power law shape of the classical models is well reproduced. Specifically, when $\phi$ reaches $\phi_m$, there is a divergence in the viscosity. This divergence is a key element of flow of mixture of grains and fluids. The fit of Eq.14 leads to a root mean squared error RMSE=5.07%. When solving a full scale debris-flow, we can expect a reasonable behaviour of the overall viscosity change when incorporating the boulders.

### 3.3 Towards full scale debris-flow modelling

Sections 3.1 and 3.2 show that the method yields accurate results for both mesoscopic and microscopic mechanics within a surge in 2D. This last section explores the flow of a $2D$ debris flow model at the surge scale with three different solid concentrations.

### 3.3.1 Geometry of the setup

Thanks to the validation of the fluid phase for complex free-surfaces and the validation of the macroscopic viscosity of the mixtures, a preliminary study of numerical experiments on debris flows in 2D is carried out.

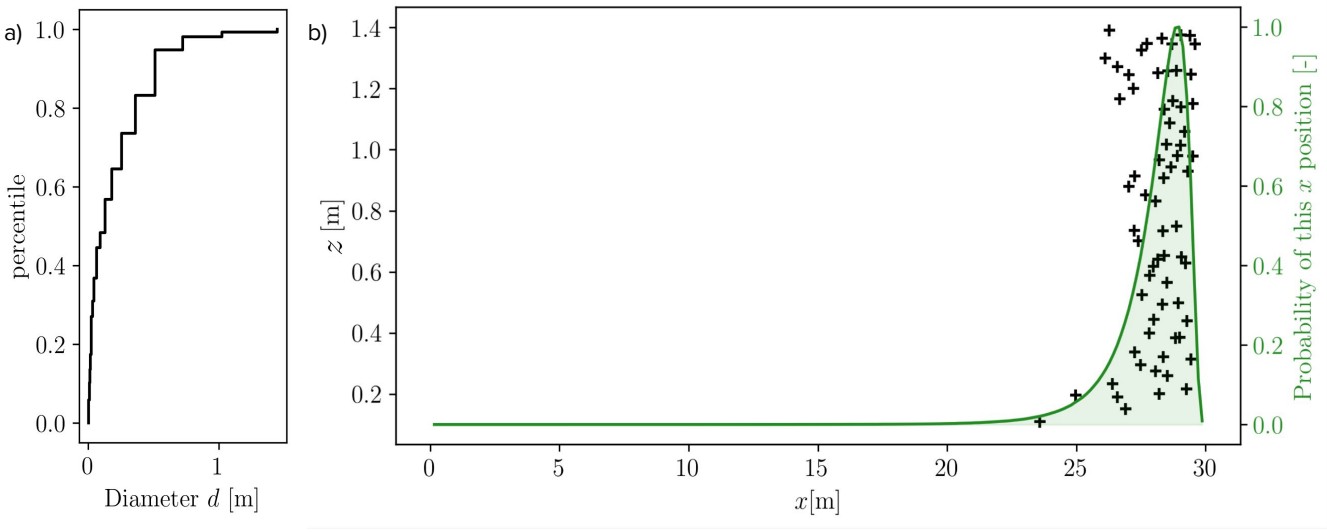

**Figure 6.** Debris flow geometry generation: a) grain size distribution measured on the debris flow event described in Piton et al. (2018), b) Example of distribution of the boulders along $x$ and $z$, the lognormal law of distribution for $x$ is illustrated in green (right axis), $z$ is random in flow depth.

Here, the setup is highly simplified to investigate the influence of boulders on the macroscopic behaviour of the flow. The numerical flow is solved in 2D using a conveyor belt channel similar to section 3.1, but to which boulders are added. The slope of the channel is taken to be similar to slopes of debris flow torrents, $S = 10.1°$.

The initial condition are based on the field data in Lapillonne et al. (2023) and Piton et al. (2018). In the following analysis, the goal is not to represent exactly a given event, but rather to use sensible orders of magnitude for the flow of interest.

The flow is initialized with constant flow depth $h_{surge} = 1.4m$ $u_b$ is then chosen so that the Froude number of the initial conditions reaches $Fr \approx 1$. The choice of this Froude number is motivated by two factors : *(i)* the flow must represent debris flow Froude numbers that are found in nature, see Lapillonne et al. (2023), *(ii)* the introduction of boulders in the flow is expected to slow down the flow, thus lowering its Froude number, compared to a grain free surge. This way, we ensure that the flow characteristics will remain in reasonable ranges compared to values of the Froude number measured on the field ($Fr < 3$, with a distribution centred around $Fr = 1$). The viscosity of the interstitial fluid is kept the same throughout all the simulations. Its value is chosen from the analytical derivation in Eq. 12 so that the grain-free flow would exhibit the features explained above ($Fr = 1$). These values are summarized in Table 3

The length of the numerical setup $L_x$ represents the front of a debris flow surge length. To determine it, the analysis was based on the volumes of the events on the upper station of the Réal torrent. The median volume of an event is $V \approx 1000m^3/s$, with a channel width of $B = 8m$. Using a simplistic rectangular approximation, this lead to a length of the flow front and body of a corresponding event $\frac{V}{h_{max}B} \approx 90m$. As the setup aims at only representing the front of the flow, the actual length of the numerical setup is $L_x = 33m$.

**Table 3.** Characteristics of the 2D debris flow experiment

| Number of boulders [-] | $\phi$ [-] | Slope [°] | $\mu_0[\mathrm{m^2\,s^{-1}}]$ | $\rho\,[kg/m^3]$ | Initial flow height [m] | $u_b[\mathrm{m/s}]$ | $d_p$ | $C_h$ |
|---|---|---|---|---|---|---|---|---|
| $[0,50,100,154]$ | $[0,0.13,0.29,0.53]$ | 10.1 | 0.280 | 2000 | 1.4 | 3.5 | 0.01 | 1.8 |

Each boulder is represented as a disk which size is randomly sampled on a grain size distribution measured both on the front and the body of an actual debris flow (see event description in Piton et al., 2018). Only grain sizes bigger than $d_{truncature} = 20cm$ and up to $h_{\max}$ are sampled. Indeed, computational time is driven partly by the number of individual grains represented, the number of collisions and the size of the pores between the grains. Thus, modelling all the singular grains in the debris flow material is not reasonable computationally. This computational barrier creates a conundrum because those small grains not only influence the overall viscosity of the interstitial fluid in which we encapsulate them, as demonstrated in the previous section; they also affect the large boulders through granular dynamics whereby small particles are driven downward and large particle migrate upward through kinetic sieving and squeeze expulsion (Gray, 2018). This phenomena is key to understand how huge boulders with a density $\approx 2600 kg/m^3$ seem to float in debris flows which average density is $\approx 1800$ to $2000 kg/m^3$. To mimic this effect that cannot be explicitly modelled due to the lack of representation of smaller grain sizes, the boulder relative density, $\frac{\rho_b}{\rho}$ where $\rho_b$ is the boulder density, was arbitrarily set to $0.9$. This crude assumption certainly deserves further investigation in future works. To place each boulder, their positions are drawn iteratively and tested to avoid any intersection with previously placed boulders. In the $z$ direction, their position is randomly drawn on a uniform probability law. In the $x$ direction, they are placed following a truncated lognormal law centred close to the front, as schematized on Fig. 6b. This means that most boulders will be generated near the front. This approach is a make shift to preferentially place boulders near the front as observed in the field (Johnson et al., 2012). The flow then reorganises their position during the simulation.

The data of interest is measured over $55s$ after the first $20s$ with a sampling period of $0.5s$. The first $20s$ of the flow are discarded because the flow has not yet reached stationarity (see Fig. S5 in the supplementary material). The time scale chosen to do the observation is of the order of magnitude of the time travel between sensors in actual field monitoring stations. The data is averaged over that duration to have a representative measurement of the flow rather than an instantaneous one. Three experiments with an increasing number of boulders are carried out and summarised in Table 3.

The value of $h_{\max}$ is taken as the maximal value of the flow height in the surge at each time-step. The free surface elevation is instantaneously very dependent on the local presence of the boulders : boulders clustering and piling up will temporarily and locally increase $h_{\max}$. The statistical variability of $h_{\max}$ was recorded to illustrate how the presence of boulders might introduce a kind of noise in this regularly monitored parameter. The upstream first $5m$ are ignored to evaluate maximal flow height in order to avoid eventual effects of the upstream wall. The Froude number and the equivalent viscosity are computed as follows :

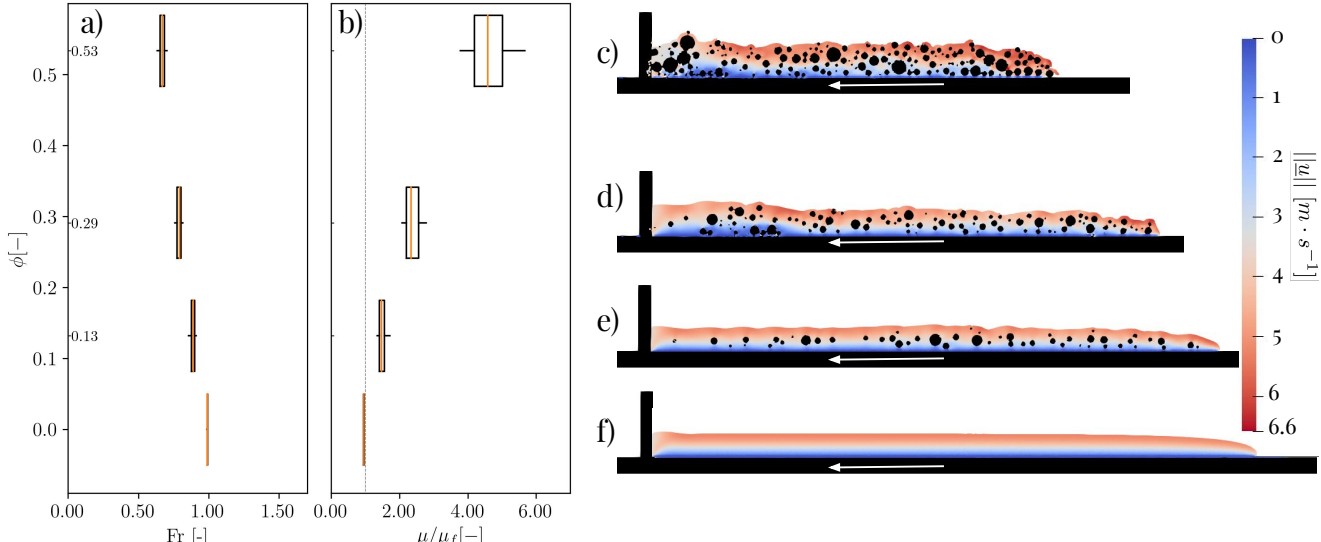

**Figure 7.** 2D debris flow model : Boxplot of a) the Froude numbers and b) the ratio of equivalent viscosity over fluid viscosity : whiskers represent the upper and lower quartiles, orange bar represents the mean value, gray vertical line on b) highlights $\mu/\mu_0 = 1$. Velocity fields and general view of the different simulations for c) $\phi = 0.53$ d) $\phi = 0.29$, e) $\phi = 0.13$ and f) $\phi = 0$. The white arrow represents the movement of the conveyor belt. Views have been tilted back to the horizontal plane in order to ease the reading of the comparison : the tilted gravity vector is shown.

$$
\begin{aligned}
Fr &= \frac{\bar{u}}{\sqrt{gh_{\max}}} \\
\mu_{eq} &= \frac{\rho g \sin\theta}{3\bar{u}} h_{\max}^2
\end{aligned}
\tag{21}
$$

### 3.3.2 Effect of the introduction of boulders on the macroscopic behaviour of the flow

On Figure 7c-f, the overall view of the surges is shown. With increasing concentration, the shape of the surge shows higher flow depth, here being a proxy of increasing apparent viscosity. Overall, the surges show a recurrent circulation of the grains throughout the simulation. The boulders stay within the flow and the simulations successfully behave as debris flow surges.

On Figure 7a, the Froude numbers of the flow are shown against the boulder concentration in boxplot form. The whiskers represent the quartiles of the sampling of $h_{max}$ over time. The Froude numbers decrease with increasing boulder concentration,

as the obstruction to the flow increases the effective viscosity of the mixture. In this Figure, the whiskers of the boxplot represent the quartiles due to the time averaging of the data. Throughout the simulation, the Froude number varies marginally as the interquartile range of the boxplot is narrow and stays $> 0.5$, i.e. in orders of magnitude consistent with the field data from (Lapillonne et al., 2023).

On Figure 7b, the viscosities are shown to increase with increasing boulder concentration. The relationship between viscosity

and boulder concentration shows a positive non-linear trend: as the concentration of grains increases, the viscosity of the

mixture exhibits a more-than-proportional growth. Both $h_{\max}$ and $\overline{u}$ are used to compute $Fr$ and $\mu$, but the latter shows stronger instantaneous variations (larger boxplot) than the first, due to the difference in non linearities regarding $h_{\max}$ in Eq. 21. These results demonstrate that complex coupled methods are a relevant tool to explore debris flow motion and internal dynamics, as well as the complementary contribution of the interstitial fluid and granular content to the overall behaviour of the surges.

## 4 Discussion

This discussion section is mirroring the organisation of section 3 so that each subsection discussed its corresponding results.

### 4.1 Viscous Newtonian surges

The accuracy of the results at different points in the flow are promising in terms of possible applications, especially close to the toe of the front. Precise knowledge about the flow front characteristics is of major interest for applications on mud-flow behaviour as well as debris flow modeling.

Overall, the method is thoroughly validated for viscous surges. Both the macroscopic behaviour and internal dynamics are recovered to a satisfactory accuracy. The errors of the simulations go well into the uncertainty brackets of the experimental measurements, meaning the validation cannot be further optimised. This is a first step towards the modelling of boulder-laden debris flows with the SPH method.

In the scope of debris flow modelling, studying the internal dynamics of these complex flow fronts will be a significant advantage to understand their global dynamics, including their flowing and stopping conditions. To do so, the validation of the internal dynamics of such a laminar viscous flow front is a necessary step to approach reasonable rendition of the internal physics. Validating some of the driving processes of the internal dynamics along with simple indicators of the macroscopic behaviour of the flow are steps forwards in modelling more realistically actual debris flows. Key elements like front velocity, flow height and front shape were correctly represented in this case study. These parameters can be measured in the field (Hürlimann et al., 2019; Lapillonne et al., 2023; Schöffl et al., 2023; Aaron et al., 2023). The most advanced monitoring stations are even equipped to measure surface velocities (Theule et al., 2017; Schöffl et al., 2023), basal forces (Nagl et al., 2020) or the 3D shape of the debris surges (Aaron et al., 2023). In further investigation, sophisticated methods of monitoring and of numerical modelling will be used in conjunction to improve our knowledge of debris flow dynamics.

Within this work, we validated this numerical approach for such applications. The knowledge on experimental data and data uncertainty for such mud-flows and debris flows in the field leads us to consider reasonable and acceptable the errors we highlighted: no model will ever be a perfect representation of such complex processes.

### 4.2 Macroscopic viscosity of mixtures

While the overall shape of the rheological response is correctly retrieved in the Poiseuille test, the actual values of the coefficient $p$ in the different fitting of empirical equations do not correspond to the usual values found in the literature. Generally, the classical fitting parameters of Krieger-Dougherty are known to not always perfectly fit the experimental observations, but to

represent broadly the behaviour of mixtures (Guazzelli and Pouliquen, 2018). However, we acknowledge that some numerical works are able to retrieve the accurate 2D descriptors of the flow such as Kromkamp et al. (2006).

The lubrication forces in SPH are highly dependent on resolution. When two solid particles interact through the fluid at close range, the lubrication forces should have a great influence on their movement. However, in SPH, these lubrication forces diverge with the decreasing number of particles. If the solid particles become too close to one another, the resolution becomes insufficient at a certain scale. To avoid such problems, some numerical methods implement an artificial additional lubrication force (Bian and Ellero, 2014; Chèvremont et al., 2020). However, in the scope of debris flow modelling, where collisions drive the macroscopic flow, this lack of precision is taken as an uncertainty of the model.

The preference of a Poiseuille setup over a classical Couette flow is not common because of the risk of drifting of the solid particle and irregular solid concentration. Irregular solid distribution due to migrations of the solid elements in a Poiseuille flume is a well known phenomenon which would lead to an incorrect estimation of the actual aerial fraction (Guazzelli and Pouliquen, 2018). In the case presented in this paper, the time scale of this rearrangement is very long compared to the time scale of stabilisation of a velocity profile. In that sense, making the assumption that no lateral migration occurs is sensible. Empirically, no migration is observed during the measurement period.

Overall, for the application of this paper, we consider that the divergence of the viscosity at increasing concentration, and the overall shape of the relationship between viscosity and concentration, is sufficient.

## 4.3 Debris flow model

Previous pioneering works did model mixtures of interstitial fluid and grains to approach debris flow behaviour, always relying on simplifying hypothesis: Canelas (2015) modelled small scale model of boulder-laden flows with the fluid as water , Leonardi et al. (2014) modelled a column collapse experiment of mud and gravel and Leonardi et al. (2015) modelled an impact of mixed grains and Bingham interstitial fluid against a flexible barrier focusing on the impact dynamics. Our present work presents proof of concept for a complex numerical model that encompasses multiple scales while being accessible for specific applications.

As explained above, the main features of the viscosity changes due to the presence of grains are reproduced by the model, i.e. its non-linearity and divergence at high concentrations (Fig. 5). For field applications, the complicated mixture of water, sand, gravel and eventual clay is assumed in a first, crude, approximation to be represented by our viscous flow. Its viscosity is thus seen as a calibration criterion that encapsulates the more complex behaviour of the mixture which precise composition is variable and unknown. The capacity of the tool to model grains is used only to capture how large boulders behave and back-influence the overall flow behaviour (Fig. 7). Eventually, the capacity to model larger surges, with more grains of eventually increasingly finer diameter is limited by the computational power.

The lack of representation of cohesive rheology via a non-Newtonian constitutive model in the SPH resolution leads to an underestimation of the plug. In Figure 7c-f, the effect of boulders on the velocity profile can be seen, with flows with a higher boulder concentration (7c-e) having a larger section with an almost constant velocity. This effectively leads to a pseudo-plug that is driven by the presence of granular matter. However, the viscous matrix should also drive a plug zone. This plug influences the shear profile in the flow and affects the relative motion of the boulders in the flowing material, impacting both

boulder mobility and spatial distribution. This assumption, combined with the 2D assumption, are acknowledged as intrinsic limitations of the model. Nonetheless, the objective of this paper is to capture macroscopic features comparable to those observed in the field for which the model remains satisfactory.

The 2D experiments we present are another proof of concept showing that coupled solid fluid numerical models are able to reproduce qualitatively the macroscopic flow features of a field scale debris-flow. However, the 2D assumption for debris-flows is unrealistic. 3D motion of boulders likely drives a lot of the efforts in the granular skeleton. Such 2D models might somewhat be comparable to a very confined modelling of the flow : the boulders not being able to move laterally. This confinement probably lead to an overestimation of the slowing down of the flow and also likely tend to under deposit grains. This confinement probably leads to an overestimation of the slowing down of the flow and also likely tends to under deposit grains, especially since it cannot represent lateral deposition and formation of levees. Moreover, there is a diminution of lubrication effects because of the 2D simplification, as the lateral rearrangement of grains are not replicated. However, we believe that the extension of this model to a 3D case is perfectly feasible: the validation of each phase would be exactly the same. Nevertheless, it is significantly more resource intensive to extend such models to 3D, and was considered out of the scope of this study.

A more complex shape of the boulders, with elements which cannot roll, would be another step toward a more realistic approach. This could be generated as an ensemble of clumped spherical elements. The interactions of these granular elements with the flat bottom would likely have an even stronger non-linear impact on the macroscopic flow. However, this first approach with simplified elements, and thus more accessible geometry, already shows the dramatic influence of boulders on the flow.

This study would benefit from a complete statistical analysis with randomly generated initial conditions and repeated occurrences. Indeed, the current result could be influenced by the initial grain distribution (size and location) sampled for the case. A statistical analysis, running this experiment with an appropriate number of initial setup for the three concentrations, would allow us to study more precise patterns such as re-circulation of boulders into the flow. However, this does not influence the validity of the model even with only one simplistic case.

The lubrication effects are under-represented not only due to resolution and dimensional effects, but also because of the nature of the interstitial fluid used in the flow. Indeed, the artificial interstitial fluid aggregates the watery composition of the overall mixture and the grain sizes ranging from clay to the cobble range ($d_{truncation} = 20cm$). This interstitial fluid should have a strong effect on the overall macroscopic rheology. The inclusion of clay and other granular material argues for the representation of the interstitial fluid as non-Newtonian with a yield criterion (Ancey, 2007). However, this simplified model already encompasses parts of the overall behaviour by incorporating granular elements and shows satisfactory features for debris flow modelling applications.

## 5   Concluding remarks

This paper shows that a solid-fluid model using DualSPHysics is reliable for debris flow modelling. The work addressed different steps which feed towards an accurate modelling of such complex flows :

- Viscous surges are validated against precise experimental data. Remarkable agreement between the simulations and the experimental data is found, with errors on the macroscopic features $< 5\%$ after convergence. The velocity profiles at three different positions in the flow are also reproduced with errors $< 5\%$ after convergence. It is, to the best of our knowledge, one of the few validation of the SPH approach for very viscous flows near the creeping threshold ($Re \approx 0.1$).

- Viscous flow loaded by small grains are investigated against the Krieger-Dougherty relationship. The overall shape of the non-linear relationship between solid concentration and macroscopic viscosity is fitted, with a root mean squared error of $5.07\%$. The divergence between the expected values of the fitted and reference coefficients driving the relationship is assumed to be due to i) the scarcity of available coefficient for $2D$ numerical models ii) a divergence in the representation of the lubrication forces at small scales within the method. The method is however still considered valid in the scope of debris flow investigation.

- 2D simplified debris flow surges with polydisperse boulders are modelled at various concentrations. Field-like behaviours were retrieved with Froude numbers in the range of the field measurements. Macroscopic viscosity is found to evolve non-linearly with the introduction of boulders. The importance of using complex models is highlighted and the model is considered promising for debris flow physics exploration. There is no argument against the extension of this model to a 3D case, which makes its potentially powerful.

To conclude, the different elements of the model are considered validated for the use of Newtonian debris flow surges. This study specifically highlights that the presence of boulders in the flow brings a non-linearity to the viscosity that should be considered when doing fully continuous fluid-like models. This model, however, is only a first stepping stone towards more complex, complete models. Specifically, encompassing the effects of non-spherical elements, or of a non-Newtonian interstitial fluid, would be of great interest to better explore debris flow physics. Nonetheless, this simplified model is able to investigate questions of interest of the debris flow community, such as the study of the dead zone when impacting a barrier or the co-interaction of the granular and flow fronts.

**Acknowledgements**

The work of S.L., V.R. and G.P. was supported by the LabEx Tec21 Investissements d'avenir - agreement n°ANR-11-LABX-0030.

**Author contribution**

Conceptualization : S.L, G.P. and V.R., Data curation : G.C., Formal analysis : S.L. and G.F., Funding acquisition : V.R. and G.P., Investigation : S.L., G.P. and G.F., Methodology : all authors, Project administration : G.P., Software : S.L. and G.F., Supervision : G.P., V.R. and G.C., Visualization : S.L., Writing – original draft : S.L., Writing – review and editing : all authors.

**Competing interest**

The authors have no competing interest to disclose.

**Code availibility**

The current version of model is available from the project website: https://dual.sphysics.org/ (GNU Lesser General Public License), LAPILLONNE, Suzanne, 2024, "Source code and parameters for viscous surges and mixtures with DualSPHysics v5.2", https://zenodo.org/records/14411481.

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
