# Peer review of "Towards viscous debris flows simulation using DualSPHysics v5.2: Internal behaviour of viscous flows and mixtures"

_EGUsphere, 2024_

## Author Response (AR1)

Dear editors,

You will find under this the response to 4 reviews including two CC.
On top of the changes highlighted in this response, we also decided to shorten the methods in the paper by moving parts of it to the supplementary material.

In hopes this finds you well
Best regards,
Suzanne LAPILLONNE, PhD
On behalf of all authors

Review 1 :

*We would like to thank Referee #1 for this thorough review. You will find in this comment the complete reply with our answers in italic type. I am also attaching a pdf of the same answers, this time in blue, if it makes it easier to read.*

The authors present a preliminary study on the routing of a viscous solid-liquid flow including boulders at the purpose of simulating the observed debris flows in the field. The study is interesting but unfortunately is not yet ready for publication. Main deficiencies are:

- The title should reflect the content of the paper: viscous debris flows or debris flow in viscous regime rather than debris flows.

*Thank you for pointing this out, it will be added to the revised manuscript.*

- At subsection 2.5 it is introduced a collision algorithm. Authors should provide some explanation about its role. In other words, it is missing some sentence where it is stated that SPH code models the viscous flow and the collision algorithm models …….? Moreover, another observation is raised about the term collision because another debris-flow rheology is the collision one (Bagnold, 1954, Takahashi, 2014).

*The collision algorithm is modelling the collisions of the boulders explicitly for section 3.2 and 3.3, but is not used in section 3.1. There are no macroscopic computations representing collision through rheology. This will be clarified in the updated document.*

- Therefore, some other explanation for explaining the difference from a collisional-rheology dominated debris flows are required.

*The debris flows of interest of this paper fall under the viscous debris flow category, as determined in Takahashi (2014), where the main driving stress in the viscous one. This will be explicitly written both in text in the introduction and the abstract.*

- About velocity profiles, it seems missing the plug, typical of cohesive rheology. Some explanation about that, it is required.

*Since we have made the assumption to use a Newtonian fluid as a first stepping stone of a more complete model, we know we are lacking the precision of the plug flow description in the pure fluid models. The investigation presented here knowingly simplifies the interstitial fluid in order to investigate the feasibility of such a numerical method to represent the complexity of a debris flow. We completely agree that using a non-Newtonian rheology would yield more accurate results in terms of velocity profile, specifically when looking at pure fluid simulations of section 3.1. We think however, that the absence of a true plug in the pure fluid models is not detrimental to the accuracy of the macroscopic behaviour of the flow, although we do acknowledge that the boulder mobility is reliant on the velocity profile. We will mention this inaccuracy in the discussion section 4.3 by adding the following paragraph :*

*"The lack of representation of cohesive rheology via a non-Newtonian constitutive model in the SPH resolution leads to an underestimation of the plug. In Figure 7c-f, the effect of boulders on the velocity profile can be seen, with flows with a higher boulder concentration*

*(7c-e) having a larger section with an almost  constant velocity. This effectively leads to a pseudo-plug that is driven by the presence of granular matter. However, the viscous matrix should also drive a plug zone. This plug influences the shear profile in the flow and affects the relative motion of the boulders in the flowing material, impacting both boulder mobility and spatial distribution.This assumption, combined with the 2D assumption, are acknowledged as intrinsic limitations of the model. Nonetheless, the objective of this paper is to capture macroscopic features comparable to those observed in the field for which the model remains satisfactory. "*

In general, some specification and explanations are missing (see the comments below)

 Some other comments as follows

Line 8 "Surges are composed of a viscous Newtonian fluid and poly-disperse boulders" This is not true at all: surges can be partially saturated and therefore dominated by friction or turbulent (Simoni et al., 2020). Authors should specify that the want to simulate a debris flow dominated by a viscous rheology. In a such a case they should introduce the dominant rheology of debris flows (see the mechanical-based classification for debris flows of Takahshi, 2014)

*We agree that we need to clarify that we are modelling debris flow which have characteristics common among european alpine streams. Both in text and in the abstract, we will add this information. In the introduction we will add "Debris flows considered in this paper fall under the viscous debris flow category (Takahashi 2014), common in the French Alps."*

Lines 11-14 "Debris flows are rapid flows saturated with non-plastic debris in a steep channel (Hungr, 2005). These fast flows yield suddenly, behaving as so-called surges, creating a granular front which has the potential to be very destructive. Debris flows evolve naturally in steep, erosion-prone catchments under intense rain as well as ice and snow melts (Recking et al., 2013)…"

The authors state that debris flows create a front without specifying anything else. After that, they mean that rain snowmelt and …. trigger debris flows. The writer suggests to specify at the beginning the cause of debris flows, in order of quantity, abundant runoff (Bernard et al., 2025); landslide (Iverson et al., 1997), snow melt and ice (Recking, 2013). After that it could added, that,  after the formation of a front, it routes downstream volumetrically growing due to the entrainment of large quantity of debris material (Reid et al., 2016; Simoni et al., 2020). Finally, it is worth to specify which type of debris flow is dominated by viscous rheology (landslide-induced debris flow for sure).

*Thank you for this comment. We will add the following precisions :*

*"Debris flows are rapid flows saturated with non-plastic debris in a steep channel (Hungr, 2005). These fast flows yield suddenly, behaving as so-called surges, creating a granular front which has the potential to be very destructive, followed by a viscous matrix engulfing granular material. Debris flows evolve naturally in steep, erosion-prone catchments and can be triggered by  abundant runoff (Bernard et al., 2025), landslides (Iverson 1997; Recking,*

Richard, and Degoutte 2013)*, or snow and ice melt* (Recking, Richard, and Degoutte 2013). *Once initiated, the flow propagates downstream, often recruiting material from the channel through entrainment* (Simoni et al. 2020)**, *(Reid, Coe, and Brien 2016)). In the European Alps, the material transported by the flow usually comes from the naturally occurring weathering processes of mountain hill-slopes, either constant or due to glaciation/ deglaciation phases* (Recking, Richard, and Degoutte 2013). *This leads to the presence of viscous dominated debris flow, with high clay content in the interstitial fluid and granular materials of a wide verity of sizes, from sand to boulders* (Coussot et al. 1998).*"*

Line 34 "However, because the models….." it should be specified these models, or the models above.

*OK, thank you, done.*

Lines 40-42  "However, depth-averaged models are only adapted to large scale representation of the flow. When dealing with the mechanical changes inside the debris flow material during the flow, they become limited as they are not designed to represent the material in depth. In addition, the large boulders that are transported by debris flows cannot be described in such models. Surge-scale-or small-scale- models aim at representing the physics of debris flow dynamics at a mesoscopic scale to better understand internal motion of the material and its interaction with infrastructures" This period is ill-posed. The depth-averaged models are 2D models and, therefore, they are approximated because they neglect the vertical exchange of momentum. For these reasons they cannot accurately simulate the transport of large boulders, the surge routing or the interaction with obstacles and structures.

*We are not completely certain we understand your comment, we offer this alternative text, which explains more explicitly the limits of the 2D assumption we wanted to stress  in the context of depth average modelling:*

*"However, depth-averaged models, by nature, are only adapted to large scale representation of the flow. As they represent the flow in 2D, they intrinsically neglect the momentum changes in depth, and thus, when dealing with the mechanical changes inside the debris flow material during the flow, they become limited. Consequently to their 2D nature, the large boulders that are transported by debris flows cannot be explicitly described in such models.*

*Conversely**, surge-scale-or  models with smaller spatial domains aim at representing the physics of debris flow dynamics at a mesoscopic and microscopic scale to better understand internal motion of the material and its interaction with infrastructures."*

*We hope this formulation is clearer and that is in line with the review vision that we share.*

Line 130 "Finally, we present the initial results to model a surge scale 2D debris-flow, investigating three different solid concentrations and their influence on the macroscopic

behaviour of the surge" This sentence, it seems to contradict what written at lines 40-45 and 97-100.

*Here, the 2D dimension refers as 2D in width, i.e. we do not take into account movements that are neither longitudinal or in the depth of the flow. Here we use 'initial' to mean the first model with this method. The sentence will be changed to "Finally, we present exploratory validated results to model a surge scale 2D debris-flow with an SPH based hybrid model, investigating three different solid concentrations and their influence on the macroscopic behaviour of the surge"*

Line 420  insert the reference Johnson et al. (2012)

*OK, thank you, done.*

Subsection 3.3.2  Authors should explain what they are modeling. The writer does not understand what panels 7c-f represent. The reproduction of a surge routing toward the right on an horizontal plane? The vertical black bar on the left what is it (the wall upstream of the channel)? What does it mean the white arrow? (the conveyor belt?).

*Figure 7c-f represents the experiments described in section 3.3.1 and Table 3. We understand the title and graphical representation in the figure make it hard to relate to the description. We will add a tilted gravity vector to show the tilted experiment better.The title of Figure 7 will be changed to :*

*2D debris flow model : Boxplot of a) the Froude numbers and b) the ratio of equivalent viscosity over fluid viscosity : whiskers represent the upper and lower quartiles, orange bar represents the mean value, gray vertical line on b) highlights $\mu/\mu_0=1$. Velocity fields and general view of the different simulations for c) $\phi=0.53$ d) $\phi = 0.29$, e) $\phi=0.13$ and f) $\phi = 0$. The white arrow represents the movement of the conveyor belt. Views have been tilted back to the horizontal plane in order to ease the reading of the comparison : the tilted gravity vector is shown.}*

Line 451-452 "The accuracy of the results at different points in the flow are promising in terms of possible applications, especially close to the toe of the front." The writer does not agree: in figure 7c-f there is not any front composed of boulders.

*This does not refer to the results of Figure 7 but of the results of section 3.1 named the same way. We will try to clarify by a short sentence which discussion section refers to which results section. We are sorry for the misunderstanding.*

**References**

Bernard, M., Barbini, M., Berti, M., Simoni, A., Boreggio, M., Gregoretti, C. (2025) Rainfall-runoff modelling in rocky headwater catchments for the prediction of debris flow occurrence. Water Resources Research Water Resources Research, 61(1), doi: 10.1029/2023WR036887

Johnson, C.G., Kokelaar, B.P., Iverson, R.M., Logan, M., LaHusen, R.G., Gray, J.M.N.T., (2012). Grain-size segregation and levee formation in geophysical mass flows. J. Geophys. Res., Earth Surf. 117. F01032

Iverson RM, Reid ME, Lahusen RG. 1997. Debris-flow mobilization from landslides. Annual Review of Earth and Planetary Sciences 25: 85–138.

Reid, M. E., Coe, J. A., and Dianne, L. B. (2016). Forecasting inundation from debris flows that grows volumetrically during travel, with application to the Oregon Coast Range, USA. Geomorphology 273, 396–411. doi: 10.1016/j.geomorph.2016.07.039

Simoni A., Bernard, M., Berti M., Boreggio M., Lanzoni S., Stancanelli L., Gregoretti C (2020) Runoff‐generated debris flows: observation of initiation conditions and erosion‐deposition dynamics along the channel at Cancia (eastern Italian Alps). Earth Surface Processes and Landforms - 45, 3556 – 3571 doi:10.1002/esp.4981

**Added references**

Bernard, M., Barbini, M., Berti, M., Simoni, A., Boreggio, M., Gregoretti, C. (2025) Rainfall-runoff modelling in rocky headwater catchments for the prediction of debris flow occurrence. Water Resources Research Water Resources Research, 61(1), doi: 10.1029/2023WR036887

Coussot, Philippe, Dominique Laigle, Massimo Arattano, Andrea Deganutti, and Lorenzo Marchi. 1998. 'Direct Determination of Rheological Characteristics of Debris Flow'. *Journal of Hydraulic Engineering* 124 (8): 865–68. https://doi.org/10.1061/(ASCE)0733-9429(1998)124:8(865).
Iverson, Richard M. 1997. 'The Physics of Debris Flows'. *Reviews of Geophysics* 35 (3): 245–96. https://doi.org/10.1029/97RG00426.
Recking, Alain, Didier Richard, and Gérard Degoutte. 2013. *Torrents et Rivières de Montagne: Dynamique et Aménagement*. Quae Editions.
Reid, Mark E., Jeffrey A. Coe, and Dianne L. Brien. 2016. 'Forecasting Inundation from Debris Flows That Grow Volumetrically during Travel, with Application to the Oregon Coast Range, USA'. *Geomorphology* 273 (November):396–411. https://doi.org/10.1016/j.geomorph.2016.07.039.
Simoni, Alessandro, Martino Bernard, Matteo Berti, Mauro Boreggio, Stefano Lanzoni, Laura Maria Stancanelli, and Carlo Gregoretti. 2020. 'Runoff-Generated Debris Flows: Observation of Initiation Conditions and Erosion–Deposition Dynamics along the Channel at Cancia (Eastern Italian Alps)'. *Earth Surface Processes and Landforms* 45 (14): 3556–71. https://doi.org/10.1002/esp.4981.

Review 2 :

*We would like to thank Reviewer #2 for this thorough review. You will find in this comment the complete reply with our answers in italic type. I am also attaching a pdf of the same answers, this time in blue, if it makes it easier to read*

1. The paper discusses several models and methods but lacks clarity in explicitly stating the gaps in the literature that the proposed model addresses. The paper should briefly elaborate on the limitations of existing models and why the coupled solid-fluid approach (e.g., SPH) is necessary. It would be helpful to discuss where existing models fall short and how your model offers a solution.

*We discuss the different existing models, and the reasoning behind the choice of SPH-CHRONO from line 74 to 113.*

*Using what we call 'hybrid models' is advantageous compared to singular mechanistic models as they describe completely the dynamics within the material (solid-solid, solid/fluid and pure fluid interactions). These types of models have only been available in the recent year because of their numerical complexity and their resource needs.*

*Hybrid models, such as the one presented here, are still relatively new in the debris flow community. Only a few have been developed and there is a need to properly validate them rigorously, which is what we offer to do in this paper. There are only a few examples of hybrid methods as of today. To our understanding, the LBM-DEM model of Leonardi et al. 2014 is the only other model that has the same applicability and underwent a similar validation process as our model. Several other applications without validation or with other limitations are cited and their limitations are pointed in the lines cited above.*

*SPH has the advantages of being relatively cheap computationally, and requiring medium expertise to use (comparatively to say, LBM, which is of higher complexity for the user).*

*The advantages of using the coupled SPH-CHRONO method shown here are :*

- *On the use of a Lagrangian method*
    - *classic Eulerian CFD struggles to represent steep free surfaces as at surge front, leading to a need to re-mesh the domain very frequently, thus having very high computational time compared to Lagrangian methods for debris flow studies*
    - *Lagrangian methods on the other hand intuitively handle the free-surface,*
- *On the choice of SPH :*
    - *SPH in particular easily handles interactions with solid objects and multiphysics scenarii,*
    - *SPH is fast when parallelized and well documented among Lagrangian methods.*

- Overall, the advantage of using SPH compared to LBM is mainly the computational cost, and accessibility of the SPH methods and softwares
- On the choice of DualSPHysics :
  - DualSPHysics is an open software that has made a lot of efforts to have accessible and easily understandable documentation as well as a very active community of users and developers.
  - It also has the advantage of having a lot of different features that can be useful when setting up cases (motion of boundaries, damping zones, etc.. ). Overall, we think these practical aspects are a huge advantage of this method when dealing with numerical studies. This helps bridging the gap between pure numerical research communities and field/experimental communities.
  - DualSPHysics implemented no slip or quasi no-slip boundary conditions that are reliable thanks to mDBC, which were a requirement to provide accurate results for high viscosity flows.
  - DualSPHysics already had multi-physics coupling (with CHRONO).
  - DualSPHysics also has a 3D implementation meaning 3D debris flow modelling will also be possible based on the validation presented here.
  - Similarly, DualSPHysics has a framework for non-Newtonian flows, meaning the addition of that complexity in the physics is easier than with other solvers.
- On the choice of CHRONO for the solid-solid interactions :
  - Using CHRONO is faster than DEM when there are many collisions.
  - In our case, DEM and CHRONO have comparable computational time because our number of collisions remain relatively low (compared to, for example, pure DEM with big size ratios between elements), but since our aim is to make the model available for other researchers having a fast solver for solid solid interaction is of interest,
  - The choice of using CHRONO in the software was motivated by practical reasons of compatibility between the versions of the software, however, there is a DEM implementation present in version 5.2 (although not compatible with mDBC),

This work focuses on validating a method for elementary mechanical processes that are of interest for the debris flow research community. Research in geophysics, including in papers published in GMD, do not systematically aim at using the "best" model for any very complex geophysical phenomenon, science also progresses when careful validation of investigation methods are performed. So evaluating whether a method does or does not perform well contributes to scientific knowledge. We do not assume this method to be relevant solely on pre-existing hypotheses: instead we want to thoroughly validate the method. We believe that taking the time and effort to properly validate the models rather than assuming good performance is important research when it comes to numerical modelling of complex flows. Our contribution aims at freeing future researchers from this difficult first step so that they can use equivalent methods for more complex real-life scenarios.

We have added after line 113 a few sentences to summarize what is discussed here.

2. While the SPH method's application to high-viscosity flow is mentioned, the paper does not sufficiently address the specific advantages of SPH over other methods, especially in terms of computational costs. The paper should include a comparison of SPH with other common methods (e.g., CFD-DEM), focusing on computational efficiency to highlight the innovative nature of the SPH method in debris flow simulations.The paper should also expand on the uniqueness of the SPH method within the broader context of debris flow simulations. For instance, it should address how SPH is suited for simulating multiphase flow, handling complex terrain, and other aspects where SPH could provide distinct advantages over other modeling techniques.

*We have answered this comment in our answer of Comment 1. It is difficult to compare computational costs directly as software and hardware implementations are vastly different, the only comparison possible would be to Leonardi et al. 2014 but their computational cost was also affected by the rheology choice. This paper does not seek to demonstrate that our model is better or faster than others. It rather seeks to verify that assuming it to realistically simulate complex flows is correct. Experience proves that such initiatives are too rare and that they help correct problems early rather than late.*

3. using DualSPHysics for simulations but does not explain why it was chosen over other potential solvers. A more in-depth justification is needed, especially a comparison with other available solvers in terms of accuracy, computational efficiency, and relevance to debris flow modeling. This would strengthen the rationale for using DualSPHysics.

*The target of the paper is to examine and provide clarity to if SPH and weakly compressible solvers are applicable to viscous flows and mixtures. Currently the DualSPHysics solver is considered to be the state-of-the-art weakly compressible multiphysics SPH solver in the community. Nevertheless, a short justification is given below for clarity.*

*The DualSPHysics software is an open-source weakly compressible solver applicable to free-surface flows, nonlinear deformation and fragmentation of the free surface capable of modelling impact flows, multiphase flows, FSI, elastic deformation of solids, non-Newtonian flows (which is the target of our future work) and many other physics which are summarised in Domínguez et al., (doi:10.1007/s40571-021-00404-2). As this paper deals with free surface flows, FSI and rheology, it makes SPH and DualSPHysics an ideal candidate for our work. Further, features such as accurate wall boundary conditions, in-build FSI and coupling with Chrono Engine project makes the solver a unique tool for simulating viscous flows and mixtures with solids and free surfaces. Alternatives such as the openFOAM VOF solvers do exist and may be applicable however, they require two phase modeling when dealing with free-surface flows that can be expensive and have a diffusive interface whereas SPH and DualSPHysics use a single phase modeling approach with a sharp interface. Other SPH open-source solvers with multiphysics are, to the best of the authors' knowledge, sparse. Accuracy for both approaches (SPH, VOF) is well documented with theoretical 2nd order convergence characteristics which are lower in practical applications and is beyond the scope of this paper. Concluding the use of a specific numerical method and/or solver depends on the problem which in this case for the aforementioned reasons makes SPH and*

*DualSPHysics suitable. This paper provides a much needed insight on the applicability of the scheme/solver to such flows which answers the reviewers question inherently.*

4.The study does not fully consider other simulation methods that could be more appropriate for debris flow modeling. A more comprehensive literature review should be conducted to include other approaches for debris flow simulations, comparing them with DualSPHysics in terms of advantages and limitations. This would help validate the choice of DualSPHysics as the most suitable method for this study.

*We have answered this comment under Comment 1, 2 and 3. .*

5. While the validation against experimental data (e.g., Freydier et al., 2017) is mentioned, the paper lacks detailed information about the experimental data's sources, accuracy, and the experimental conditions. Including more specifics about the experimental setup, equipment, and conditions would strengthen the credibility of the validation process.

*These experiments were not conducted as part of the study but were the object for two publications, one in Journal of non-Newtonian Fluid Mechanics (Freydier et al. 2017) and one in Journal of Fluid Mechanics (Chambon et al. 2020). Experiments are thoroughly described in Freydier et al. 2017 who kindly provided the data to us to reproduce their experiments. Our paper being quite long and this dataset being only one of the several validation tests we perform, we believe that we cannot go into fine details about how the experimental setup: it can anyway be found in the original papers.*

*We however understand that the readers will be interested by a few more details and we added the following sentences precision (in bold below):*

*\cite{freydier_experimental_2017} present experimental data precisely measuring viscous surges**, i.e. flowing material, close from the maximum depth, rushing over a dry bed. Using advanced velocity measurements techniques by image analysis of a seeded, transparent fluids enlighten with a longitudinal, vertical laser sheet**, these experiments captured both the macroscopic behaviour of a viscous flow front - monitoring properties of the flow such as free surface elevation **and front velocity** - and internal dynamics within the flow front - measuring velocity profiles. The experimental setup is composed a conveyor belt tilted to a chosen angle, transparent side-walls, and a wall upstream the conveyor belt forcing a flow front to form steadily. This experimental setup is thoroughly described in \cite{freydier_experimental_2017} and \cite{CHAMBON200954}.*

*Fluids used are transparent mixtures of glucose and water allowing to measure accurate velocity fields and free surface profiles. \cite{freydthese} showed that these flows have a viscosity independent from strain and strain-rate, varying with concentration of glucose in the mixture. High definition, high velocity images at different location of the flow and complete velocimetry within the surge provide a full description of both the free surface shape and the velocity fields within the flow.*

*The characteristics of the experiments used in this current work for numerical validation of the software are shown in Table \ref{tab:charac_exp}.*

6.The study presents simulations of a simplified debris flow but does not discuss the model's limitations, particularly in terms of real-world applicability. A discussion of how the model handles complex terrain, non-Newtonian fluids, and other practical challenges would be valuable. Moreover, suggestions for improving the model to account for these complexities would provide valuable insights.

*This point is raised in the discussion line 504 to 529. We discuss the limitation of the model for real world applications and the future developments envisioned. We have started exploring non-Newtonian rheologies with the same method, for which preliminary results can be seen in Lapillonne (2024). Technical difficulties highlighted in the manuscript are hoped to be overcome in the future.*

*Lapillonne S.. Modelling debris flow surges with a coupled solid-fluid model : a multi-scale investigation. Fluid mechanics [physics.class-ph]. Université Grenoble Alpes [2020-..], 2024. English. ⟨NNT : 2024GRALI034⟩. ⟨tel-04716855⟩, available here : https://theses.hal.science/tel-04716855/*

7.While various fluid and granular models are mentioned, the paper should critically evaluate the limitations of these models and explain why SPH provides a clear improvement over them. More in-depth discussion is needed regarding how SPH can address specific challenges that other models cannot, especially in terms of scalability, accuracy, and complex flow dynamics.

*We are not sure we understand what this comment specifically addresses, we have provided an extensive explanation under Comment 1.*

8.  The discussion on how solid concentration affects flow behavior is interesting but lacks sufficient detail. Providing more results or graphs showing how changes in solid concentration influence the flow dynamics would help to better understand this relationship. Additionally, it would be helpful to compare these model results with field observations or experimental data to validate the findings.

*Section 3.2/4.2 reflect on the ability of the model to represent the macroscopic changes in the viscosity. The effect of solid concentration on viscosity is a known phenomenon in suspension studies (see Stickel and Powel 2005, Boyer et al 2011 and Mewis and Wagner 2012 among many others for example) and are not the focus of these sections. Here we just intend to show that the non-linear relationship between concentration and viscosity is retrieved by matching it to a well-known, well used power law which is experimentally derived : Krieger Dougherty ,1959.*

*Stickel, J. J., & Powell, R. L. (2005). Fluid mechanics and rheology of dense suspensions. Annu. Rev. Fluid Mech., 37(1), 129-149*

*Boyer, F., Guazzelli, É., & Pouliquen, O. (2011). Unifying suspension and granular rheology. Physical review letters, 107(18), 188301.*

*Mewis, J., & Wagner, N. J. (2012). Colloidal suspension rheology. Cambridge university press.*

*Section 3.3/4.3 does not aim at demonstrating the exact change. We believe that showing quantitatively the relationship between macroscopic viscosity and boulder concentration would require a thorough parametric study, with larger size ratio between the boulders. This was deemed out of the scope of a validation paper but would be interesting for a study focusing on that particular effect. Here the aim is to show that the numerical method renders correct overall behaviour and that it seems that the associated effects could be key in the driving of the overall surge viscosity, opening new prospects for further researchers.*

9. In the conclusion, the potential contributions of the model to debris flow research are highlighted, but the discussion could be more compelling. To make the contributions clearer, consider framing them in terms of specific research challenges that the model addresses. For example, how does the work advance our understanding of solid-fluid interactions or improve simulations of complex debris flow behavior compared to existing models?

*The aim of the paper is written in the abstract and the introduction : "This paper investigates the accuracy of a solid-fluid model using the SPH software DualSPHysics v5.2 coupled with ProjectChrono for debris flow modelling. It focuses on different validation steps of the method, both for pure fluid and a mixture of fluid and boulders to build reliability of the model to prepare for the simulation of a simplified debris flow." This paper is not meant to advance our understanding of solid-fluid interactions but rather to show how numerical models can perform in the context of debris flow modelling. As explained in the paper, debris flow numerical models are scarce, there is a need to explore the accuracy of hybrid models before application. To our knowledge, the only comparable model is the LBM-DEM approach in Leonardi et al. 2014. Our method is an alternative path that has the advantage of being easier to use for new modellers. We believe however that both models have their own place in the debris flow community.*

Community comment :
*We would like to thank Andrew Mitchell, for this thorough review and the interest granted to our preprint. You will find in this comment the complete reply with our answers in italic type. I am also attaching a pdf of the same answers, this time in blue, if it makes it easier to read.*

Overview:

The manuscript provides a detailed examination of flow-surge behaviour for both viscous fluid, and a mixture of viscous fluid with solid particles. The coupled SPH and solid particle modelling was used to replicate two lab experiments, and recover velocity and free-surface profiles. A simplified simulation of a single debris flow surge with poly-disperse boulders was also carried out, yielding realistic Froude numbers for the flow.

General comments:

1. It is unclear to me if the first part of the work focused on a viscous fluid without solid particles is applied to the part of the work with solid particles. Are these two independent model validations, as a lower viscosity is used in the cases where particles are present?

*The lower viscosity is used in the 1st case due to scaling effects. We wanted mainly to answer the question : can SPH, traditionally used for medium to high Reynolds number flows, be used for slow laminar flow dominated by the viscous regime? Since the answer is shown to be positive, we apply the certainty that the fluid mechanics part of the viscous regime behaves accurately when we model a larger and more complex flow with debris. This is why we also take a side step to ensure that the addition of boulders to the fluid mechanics method yields accurate changes in the viscosity of the overall flow. Then the application of the debris flow model is reliable because each technical 'brick' has been proven to behave correctly. We could have had perfectly acceptable Froude numbers with a method that inaccurately represents creeping flows. Here we ensure that both mesoscopic and microscopic scales behave accurately.*

*This will be clarified a bit more precisely in the end of the introduction to explain the relationship between each section.*

2. The flow of the paper may be improved if Section 2 started with the description of the SPH method and collision algorithm, then describe the two datasets used for model evaluation (Section 2.4, 2.5, 2.1, 2.2 and 2.3 for a suggested order). That way the information on the specifics of each comparison dataset are fresher in the readers' mind going into Section 3.

*We agree, thank you for this excellent suggestion. This will be done in the revised manuscript.*

Specific comments:

Line 22: Replace "understand" with "replicate".

*OK, thank you, done.*

Line 27: "to better understand hazard mapping" – models are typically used as an input to develop hazard maps.

*Yes, we will change the text to "Such models are typically used as an input for the design of hazard maps on a debris fan."*

Lines 27 – 28: Depth-averaged models should be clearly linked to event-scale models.

*We will add "Event scale models rely of depth averaged methods, where the flow is assumed to be 2D, simplified in depth. Depth averaged models can either be single phase …"*

Line 42: Word choice with "in depth" – there may be confusion when discussing depth as in depth of material versus level of detail. This could be re-worded to "… are not designed to represent all the details of the internal mechanics of the flow."

*Thank you for pointing this out, we will change the text to your suggestion.*

Line 56: Muddy debris flows are not introduced previously, it would be helpful to provide some background on muddy versus granular debris flows.

*You are correct. With the changes done from the comments of reviewer 1, we will clarify a bit better that in the context of the european alps, we consider viscous-driven muddy debris flows in this paper. In this section we will change to "As a first approximation, muddy debris-flows, which are mechanically driven by viscous motion, can be studied at the macroscopic scale…"*

Line 81: Replace "inconvenient" with "inconvenience".

*OK, thank you, done.*

Lines 100 – 102: This sentence is out of place, move to line 105 or after to keep the information on previous work together.

*OK, thank you, done.*

Line 130: Provide a reference for the Krieger-Dougherty equation.

*OK, thank you, done, added (Krieger and Dougherty 1959).*

Krieger, Irvin M., and Thomas J. Dougherty. 1959. 'A Mechanism for Non‑Newtonian Flow in Suspensions of Rigid Spheres'. *Transactions of the Society of Rheology* 3 (1): 137–52. https://doi.org/10.1122/1.548848.

Line 135: The paper deals with significantly more than viscous laminar surges.

*Reformulated to*

*"The aim of this paper is to model viscous laminar debris flow surges with Reynolds number close to the creeping flow limits ($Re \approx 0.1$). "*

Section 2.4.1: It would be helpful to provide a brief description of the SPH method (i.e., discrete particles with a free surface interpolated from particle interactions). That would help with the discussion of the smoothing kernel for readers unfamiliar with SPH.

*We will add between 2.4 and 2.4.1 : "Smoothed Particles Hydrodynamics is a computational fluid dynamics method based on the lagrangian framework. In SPH, the continuous domain is discretized into numerical nodes ('particles'), which are points of known information. Typical properties of the continuum (e.g. velocity, density, … ) are associated to each of these points. The SPH method relies on the resolution of the Navier Stokes equation via interpolation onto these nodes. Particles interact with each other in a defined neighbourhood, named a smoothing kernel, by resolving the Navier Stokes equation system."*

Lines 301 – 308: These two paragraphs seem to be methodology for the numerical study as opposed to results.

*This is the presentation of the setup, we decided to include it in the results section so as to not divide the paper into three sections Setup / Results / Discussion with each repeating the three experiments. I agree that this could be moved to methodology, but we believe it will be harder to follow for the reader, as the convergence study is a crucial step of our investigation.*

Lines 311 – 312: Could you provide some description or a reference for "parasite fluctuations"?

*Yes, we will add : "With SPH, velocity of the continuum can only be estimated through the velocity of the particles in the discretized flow. With the movement of particles in the flow intrinsic to the SPH method, instantaneous measurements can be parasited by fluctuations in the positions of the particles within the sampling window. To avoid these instantaneous effects, the results are averaged over 5 seconds."*

Figure 2: It is not clear what the optimum value from the convergence study is. From Tables 2 and 3, I can infer that 1.8 was selected for $C_h$, but it would be good to explicit state that choice in the discussion of this figure (similar comment for Figure 3).

*In both cases, Ch 1.8 has a good convergence, with precision increasing or capping with decreasing value of dp. The other candidate to have a similar convergence is 2.2, however 2.2 leads to higher computational times, thus 1.8 was selected. This will be explained at the end of section 3.1 as a concluding sentence.*

Lines 345 – 346: Repetition with "close to the free-surface" and "especially near the free-surface".

*OK thank you, corrected*

Lines 359 – 360: The statement "guarantee the correct rendering" is very definitive statement, and I would argue all models, no matter how sophisticated are significantly simpler than a real debris flow. I would suggest changing the wording to "develop a more accurate representation".

*OK thank you, corrected*

Lines 408 – 410: This sentence is unclear and should be revised, I think the main point is that modelling each individual grain is not feasible computationally.

*Reformulated to "Indeed, computational time is driven partly by the number of individual grains represented, the number of collisions and the size of the pores between the grains. Thus, modelling all the singular grains in the debris flow material is not reasonable computationally."*

Line 416: Given, as you say, the assumption of boulder relative density of 0.9 was arbitrary, was there any work done on testing this assumption? Using this relative density assumption is a very interesting method to represent the effect of boulders being supported on the smaller particles within a flow in a computationally efficient way.

*Yes, values of relative density of 0.8,1 and 1.1 were also tested. We initially thought we would use a relative density of 1 but values of 1 and 1.1 did not perform as well as 0.9. This was difficult to quantify in a readable way, and it is more of a qualitative statement. In the end, our choice was a bit arbitrary so this step is not presented. Overall, there is improvement to be made to really understand and methodically explore this idea but it would be out of the scope of this paper.*

Line 461: Wording around "ensuring features will be rendered correctly" is also very definitive. In my mind there is still a jump between the very detailed work done in this paper and having confidence that we have correctly accounted for all of the chaotic elements within a debris flow. This is in no way a criticism of the work done, which provides a very thorough testing of the numerical method, but I do think some of the more definite statements should be softened to reflect the uncertainty and approximations of any modelling study.

*Yes we agree, we think the sentence is misleading : we represent very well our model material, our 'mockup' debris flow, but because we don't represent the whole complexity of the debris flow material, this is too definitive. We will rephrase to " Validating some of the driving processes of the internal dynamics along with simple indicators of the macroscopic behaviour of the flow are steps forwards in modelling more realistically actual debris flows."*

Line 500: "viscosity is thus seen as a calibration parameter" – it was not clear previously that the viscosity was calibrated on a case-by-case basis.

*Sorry this sentence is misleading, we meant that the overall viscosity of the surge is what was used as a criterion to validate the model. Reformulated to "viscosity is thus seen as a calibration criterion".*

Line 507: Replace "no" with "not".

*OK thank you, corrected*

Lines 507 – 508: Another feature to note is the lateral movement of boulders leading to levee formation which isn't captured in the 2D model.

*OK thank you, added :*

*"This confinement probably leads to an overestimation of the slowing down of the flow and also likely tends to under deposit grains, especially since it cannot represent lateral deposition and formation of levees. Moreover, there is a diminution of …."*

Line 535: The creeping threshold is stated as ~ 1, but on line 135, 0.1 is given.

*OK thank you, corrected to 0.1.*

Line 546: Replace "potential" with "potentially".

*OK thank you, corrected*

Community comment
*We would like to thank Chan Young Yune for this thorough review and the interest granted to our preprint. You will find in this comment the complete reply with our answers in italic type. I am also attaching a pdf of the same answers, this time in blue, if it makes it easier to read.*

The manuscript deals with interesting simulations of viscous flows and mixtures with grains using DualSPHysics. I have a couple of comments as follows.

1. "Introduction" includes comprehensive explanations of the related works but is also too lengthy. It needs to be deliberately and concisely reorganized. For example, modelling with CFD-DEM appeared two times in lines 77 to 85 and 103 to 104.

*OK thank you for this comment, we will shorten the introduction to make it more concise.*

1. What is the advantage of the analysis technique used in the study over CFD-DEM? If the advantages using DualSPHysics incorporating with collision algorithm is stated in the manuscript, it will be helpful for other researchers to employ similar approach.

*The advantages of using the method shown here are :*

- *classic Eulerian CFD struggles to represent steep free surfaces as at surge front, leading to a need to re-mesh the domain very frequently, thus having very high computational time compared to Lagrangian methods for debris flow studies*
- *SPH is fast when parallelized and well documented among Lagrangian methods. Overall, the advantage of using SPH compared to LBM is mainly the computational time, and accessibility of the SPH methods and softwares*
- *Using CHRONO is also faster than DEM when there are many collisions. In our case, DEM and CHRONO have comparable computational time because our number of collisions remain relatively low (compared to, for example, pure DEM with big size ratios between elements). The choice of using CHRONO in the software was mainly motivated by practical reasons of compatibility between the versions of the software, however, there is a DEM implementation present in version 5.2 (although not compatible with mDBC),*
- *DualSPHysics is an open software that has made a lot of efforts to have accessible and easily understandable documentation as well as a very active community of users and developers. It also has the advantage to have a lot of different features that can be useful when setting up cases (motion of boundaries, damping zones, etc.. ). Overall, we think these practical aspects are a huge advantage of this method when dealing with numerical studies. This helps bridging the gap between pure numerical research communities and field/experimental communities.*

*In the end, since hybrid methods are relatively new, we think exploration of different methods is still needed to determine what would be best. However, DualSPHysics is a very strong contender because of its accessibility and its relatively low computational times. Now that*

*this method is validated, we hope to see it being applied to wider, more complex scenarii. We will add a few sentences on such advantages around line 125.*

1. Is this necessary to validate the simulation twice with viscous fluid without solid particles and again fluid with solid particles considering field conditions? I think it makes the manuscript too lengthy as well. Isn't it better to shorten the first validation and explain logically to have more relevance to the second validation?

*The validation of the pure fluid simulation does bear the weight of validating the behaviour of creeping flows for SPH and DualSPHysics against experimental data. It is quite lengthy, but this is also of use for the SPH community because, to the best of our knowledge, it has not been done. In the method section, it's important to ensure that the numerical method for the fluid mechanics part actually renders results that are expected for such flow. We really believe it is necessary to validate each technical 'package' separately, and we want to encourage the community to do the same. We agree that the paper is long. We thought it better to present such a comprehensive study rather than do salami-slicing and publish several papers not showing the broad view of performing the type of modelling we intend to perform. We will try to reframe the way these two sections are articulated so that we can shorten the first section.*

1. The study simplifies debris flow behavior by representing it as a combination of a viscous Newtonian fluid and poly-disperse solid particles. However, this neglects the complex non-Newtonian characteristics of real debris flows, such as yield stress behavior and inter-particle dynamics. Debris flow has inertia and frictional behavior between particles during the flow process. What is the authors' opinion on the limitations of this simulation considering this?

*We agree that this complexity is lacking in the model, as pointed out in section 4.3. One of the reasons we named the paper 'Towards … ' was to highlight that these validation steps are crucial for a complete numerical model to be built but are not the final step. Any model is a simplified vision of reality, here we both simplify the fluid and the granular content, as pointed out in the discussion. Many authors also use a Newtonian fluid hypothesis in complex flow modelling but do not focus on validating the code on actual measurement of viscous Newtonian flows. By doing so, we attract the attention on this point, but we validate that the code is correctly behaving against experiments when using this assumption. Many codes are published and used without heavy, thorough validation. We believe doing so is good scientific practice even though it is a bit lengthy.*

*Our hope is that this validation will free further studies from these first arduous steps and will allow them to incorporate non-Newtonian rheology into the model, without having to worry about the feasibility of using this method in the context of slow creeping flows as in debris flow research. Our opinion is that these do still represent macroscopically the flow in a way that can be used for some studies, e.g. where the values of the shear stress gradient and the plug flow does not directly impact the object of interest  (for example, entrainment ) and are simplified enough to be usable by practitioners. Non-Newtonian rheologies are a crucial*

*next step, but they do require much higher complexity and computational time. We have started exploring non-Newtonian rheologies with the same method, for which preliminary results can be seen in Lapillonne (2024). Technical difficulties highlighted in the manuscript are hoped to be overcome in the future to then compare Newtonian and non-Newtonian models and answer : how much complexity is needed to accurately represent the motion of the flow ?*

*Lapillonne S.. Modelling debris flow surges with a coupled solid-fluid model : a multi-scale investigation. Fluid mechanics [physics.class-ph]. Université Grenoble Alpes [2020-..], 2024. English. ⟨NNT : 2024GRALI034⟩. ⟨tel-04716855⟩, available here : https://theses.hal.science/tel-04716855/*

There are minor comments on this manuscript as well.

1. In line 67, change "the estimation impact forces" as "the estimation of impact forces"

*OK thank you, corrected*

2. In line 81, change "Due to this technical inconvenient" as "Due to this technical inconvenience"

*OK thank you, corrected*

3. In line 88, change "Their study contributed to understanding" as "Their study contributed to understand".

*OK thank you, corrected*

4. In line 91, change "both a pure fluid phase and a soil phase" as "both a pure fluid phase and a solid phase"

*OK thank you, corrected*

5. In line 167, change "work by (Einstein, 1906)" as "work by Einstein (1906)"

*OK thank you, corrected*

6. In line 170, change "was extended to any dimension D by (Brady, 1983)" as "was extended to any dimension D by Brady (1983)"

*OK thank you, corrected*

7. In line 177, change "at high volumetric fractions Guazzelli and Pouliquen (2018)" as "at high volumetric fractions (Guazzelli and Pouliquen, 2018)"

*OK thank you, corrected*

8. In line 201, change "to substitute the the value of" as "to substitute the value of"

*OK thank you, corrected*

9. In the legend of Figure 1, solid line should be changed as dashed line.

*OK thank you, corrected*

10. In line 301, what is kernel coefficients? Adding physical or mathematical meaning of this coefficient will be helpful for the understanding of readers.

*They are defined line 224. We will add "Kernel coefficient are a measure of the ratio between smoothing length and particle spacing" and remind the reader of the mathematical definition.*

11. In line 414, change "which average density is ≈ 1800−−2000kg/m³" as change "with average density of about 1800 to 2000kg/m³"

*OK thank you, corrected*

12. In Figure 7 (a), the Froude numbers are less than 1 and mostly lie between 0.5 to 0.8 even though, in lines 392 to 399, the value showed a distribution centred around 1. Is there a specific reason or didn't you need to change simulation conditions to show the value similar to this?

*In the field, the values are centered around 1 and span from 0.5 to 3. Here we want to be in the same order of magnitude so we start from 1 and the Froude number decreased with the progressive addition of boulders. We chose not to start from a supercritical regime for simplicity of the intercomparison between all the chosen setup, since it could have led to difficulties to compare between the sub- and supercritical cases.*